# Multilayer modelling of waves generated by explosive subaqueous volcanism

Matthew W. Hayward[1], Colin N. Whittaker[1], Emily M. Lane[2], William L. Power[3], Stéphane Popinet[4], and James D. L. White[5]

[1]Civil and Environmental Engineering, University of Auckland, New Zealand
[2]NIWA Taihoro Nukurangi, Christchurch, New Zealand
[3]GNS Science Te Pū Ao, Wellington, New Zealand
[4]Institut Jean le Rond d'Alembert, Sorbonne Université, CNRS, Paris, France
[5]Geology Department, University of Otago, Dunedin, New Zealand

**Correspondence:** Matthew Hayward (m.hayward@auckland.ac.nz)

**Abstract.** Theoretical source models of underwater explosions are often applied in studying tsunami hazards associated with subaqueous volcanism; however, their use in numerical codes based on the shallow water equations can neglect the significant dispersion of the generated wavefield. A non-hydrostatic multilayer method is validated against a laboratory-scale experiment of wave generation from instantaneous disturbances and at field-scale subaqueous explosions at Mono Lake, California, utilising the relevant theoretical models. The numerical method accurately reproduces the range of observed wave characteristics for positive disturbances and suggests a relationship of extended initial troughs for negative disturbances at low dispersivity and high nonlinearity parameters. Satisfactory amplitudes and phase velocities within the initial wave group are found using underwater explosion models at Mono Lake. The scheme is then applied to modelling tsunamis generated by volcanic explosions at Lake Taupō, New Zealand, for a magnitude representing an ejecta volume of 0.1 km$^3$. Waves reach all shores within 15 minutes with maximum incident crest amplitudes around 0.2 m at shores near the source. This work shows that the multilayer scheme used is computationally efficient and able to capture a wide range of wave characteristics, including dispersive effects, which is necessary when investigating subaqueous explosions. This research therefore provides the foundation for future studies involving a rigorous probabilistic hazard assessment to quantify the risks and relative significance of this tsunami source mechanism.

## 1 Introduction

Subaqueous eruptions are poorly understood volcanic phenomena that can generate hazardous tsunamis. When fully submerged, eruptions can transfer energy into generating impulsive waves by the displacement of water, either by regular or periodic jetting, flank instability or explosion (Duffy, 1992; Egorov, 2007; Paris et al., 2014). These processes can expand the hazard footprint of an eruption far beyond the classical primary hazards, posing danger along the shores of caldera lakes and coastal areas near explosive volcanoes. While volcanoes are estimated to be responsible for just 5% of all noted tsunamis since 1600 AD, they can be particularly dangerous in that they account for 20-25% of all recorded fatalities resulting from volcanic

activity (Mastin and Witter, 2000). However, tsunamis are often neglected from hazard maps of volcanoes. Recent events such as at Anak Krakatau in late 2018 have underlined the need to consider them in any disaster risk response (Grilli et al., 2019; Williams et al., 2019; Ye et al., 2020). This is especially pertinent for communities that are not exposed to or are less familiar

with seismogenic tsunamis and may not consider themselves at risk.

While experimental data and detailed field studies of these phenomena are rare, some reliable observations exist. For example, at Karymskoye Lake, 1996, a Surtseyan-style eruption was partially witnessed from the air including six explosions followed by tsunamis and base surges. Later ground investigation revealed run-up along the lake ranging from 19 – 1.8 m at distances 0.5 – 3 km from the vent, debris flows down the Karymskaya River, and boulder transportation up to 60 m inland

(Belousov et al., 2000; Torsvik et al., 2010; Ulvrová et al., 2014; Falvard et al., 2018). The Ritter Island volcano generated one of the largest known tsunamigenic flank collapses in 1888, leaving only a small remnant above the water surface, and has since experienced occasional submarine eruptive activity and small local tsunamis in 1972, 1974 and 2007 (Johnson, 1987; Dondin et al., 2012). In the Caribbean near Grenada, Kick'em Jenny volcano was discovered mid-eruption in 1939 and has been regularly active and progressively shoaling, with eruption columns breaching the surface in 1939 and 1965 generating

minor tsunami waves (Smith and Shepherd, 1993, 1996). Many other candidate eruptions are historically documented with small amplitude waves such as Kavachi, Solomon Islands, or lack detailed proximal observations, which leaves uncertainty as to the source mechanism responsible, for example, the 1952 Myojin-Sho submarine eruption which destroyed a naval research vessel, or the 1883 eruption of Krakatau (Dietz and Sheehy, 1954; Nomanbhoy and Satake, 1995).

Explosive volcanic eruptions are characterised by a directional gas-driven jet from the source, release of water vapour

and, in subaqueous settings, potentially violent vaporisation of sea or lake water on interaction with hot magma. This violent vaporisation is a characteristic of phreatomagmatic eruptions, and leads to rapid expansion of the resultant water vapour at depth leading to disturbance of the water surface and propagation of waves. To investigate the potential hazard range, we need to understand the relationship between source parameters of the eruption and the nature of waves they generate.

Underwater explosions are well documented (Cole, 1948; Le Méhauté, 1971; Mirchina and Pelinovsky, 1988; Kedrinskiy,

2006; Egorov, 2007) primarily owing to military reports and research in blast mitigation and structural response in, for example, ship hulls and other coastal or off-shore structures (Klaseboer et al., 2005; Aman et al., 2012; Liu et al., 2018). As a result, significant research efforts have usually been focused on non-linear fluid-structure interactions such as pressure loading from shock waves rather than any wave generation relationships. Still, some tests were conducted on this matter during the nuclear-testing age and led to the development of theoretical models describing explosion-surface interaction and dynamics of the

resultant wave field (Le Méhauté and Wang, 1996). Physical experimentation since the end of nuclear testing has been rare due to cost, practicalities, environmental concerns and the challenges of scale experienced by previous tests. In their place, numerical investigations are now the predominant area of research and offer the most likely route to advance the understanding of these processes.

The current theoretical models summarised by Le Méhauté and Wang (1996) have been used in recent years to simulate

the wavefield generated from events that produce analogous water surface cavitation such as subaqueous volcanic explosions (Torsvik et al., 2010; Ulvrová et al., 2014; Paris et al., 2019; Paris and Ulvrová, 2019) and asteroids impacting in ocean

basins (Ward and Asphaug, 2000). However, numerical solutions often either utilise the empirically derived relations without validating their use in a numerical scheme against a suitable explosive physical experiment or test a generation mechanism in the local spatial range only at the cost of neglecting investigation of the generated wave field. Often, models such as those

based on non-linear shallow water equations are applied to these problems without considering how dispersive the resultant waves may be (Paris and Ulvrová, 2019). Non-hydrostatic (NH) multilayer models such as NHWAVE (Ma et al., 2012) and SWASH (Zijlema et al., 2011) have been developed over the last 15 years to allow a wide range of spatial scales of flows such as surface waves in complex environments to be simulated in the same framework, and have been applied to dispersive and nonlinear tsunamis (Ma et al., 2013; Grilli et al., 2015; Glasbergen, 2018; Schambach et al., 2019; Ruffini et al., 2019; Grilli

et al., 2019, 2021; Schambach et al., 2021).

This work uses a recently developed non-hydrostatic multilayer solver for free-surface flows to model the physical problem. Firstly, the method is validated against a laboratory-scale experiment of released columns of water to ascertain the numerical solution's robustness in resolving a simplified comparable wave generation mechanism. Secondly, data from one of the last military explosive test series focused on surface wave observations is compared with results produced by implementing the

theoretical model's initial conditions in the numerical method. These tests are to establish fitness of the underlying models, which are then applied to a hypothetical explosive subaqueous eruption at Lake Taupō, New Zealand to provide an example of how these models can be used.

## 2    Methodology

### 2.1    Underwater eruption model

An explosive subaqueous volcanic eruption is a dynamic and complex event involving abrupt fragmentation, volume change and numerous high energy interactions between pressurised magma, volatiles and water. Their wave generation capability depends on numerous physical parameters, including eruptive energy, depth, duration, and vent geometry (Egorov, 2007; Paris, 2015). Scarce availability of field observations combined with practical limitations both in field and laboratory necessitates simplifications to be made for an explosive eruption model such as considering it as a point-source explosion, as proposed and

utilised by Torsvik et al. (2010), Ulvrová et al. (2014), and Paris and Ulvrová (2019).

The models developed for subaqueous explosions and their waves are derived from experimental data and visual observations from chemical and nuclear explosive testing during the 20th century. As documented at the time, water disturbances are born from the generation and rapid expansion of a gas bubble that interacts with the free-surface by collapsing into a crater-like cavity, accompanied by central jets of water and the initial dissipative cylindrical bore, which radially expands outward. The

resultant cavity rapidly fills under gravity to produce the second, larger jet which produces a further cylindrical bore, after which the disturbance oscillates until rest, precipitating waves of decreasing amplitude. This free-surface interaction is strongly linked with the depth of explosion relative to its energy; small-yield or deep detonations lead the explosive bubble to transfer a large portion of its energy to the surrounding water through rapid oscillations and significantly reduce wave-making efficiency. (Le Méhauté, 1971; Le Méhauté and Wang, 1996)

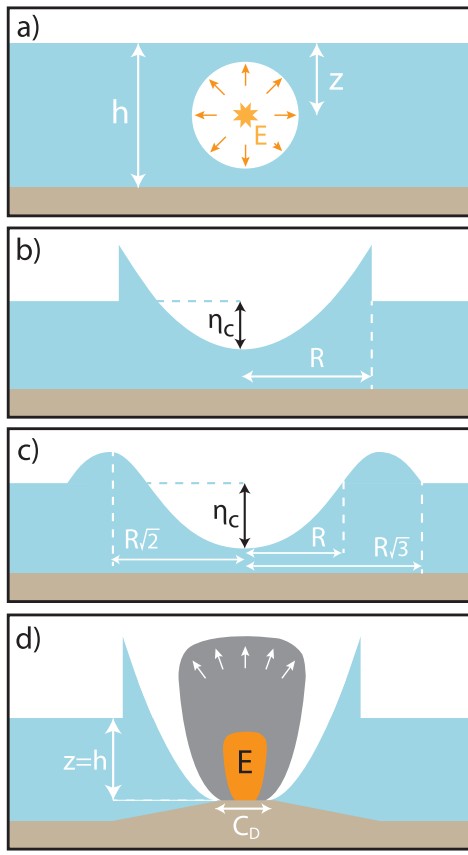

**Figure 1.** Illustrations of the subaqueous explosion problem. (a) An explosion of yield $E$ at depth $z$ in water of depth $h$. (b) The initial surface elevation profile is given by Eq. (1). (c) The initial surface elevation profile is given by Eq. (2). (d) Schematic of a volcanic scenario where such an explosion would occur at maximum depth where $z = h$, and crater diameter $C_D$ can be measured or calculated using estimated ejecta volume $V$ with Eqs. (9-10).

Bubble dynamics is a very active area of research in computational fluid dynamics (CFD), though, in the explosive realm, the focus is usually on pressure waves and solid interactions (Wang et al., 2018). These studies are usually short in temporal range and are very computationally expensive as modelling the full problem requires accounting for compressibility and multiphase flow; thus, researchers have generated specialised codes as a solution (Hallquist, 1994; Li et al., 2018). Only in recent years have studies directly simulated expanding explosive bubbles interacting with deformable beds and a free-surface (Petrov and Schmidt, 2015; Daramizadeh and Ansari, 2015; Xu et al., 2020). However, there is minimal focus on subsequent surface waves, let alone relations tested for their generation mechanisms or far-field propagation. Recently, physical experiments that utilise underwater gas jets by Shen et al. (2021a, b, c) quantify wave generation regimes and establish relationships between jet intensity, duration, depth and maximum wave height, showing that for a given jet intensity, there is a critical depth and minimum duration limit that produces maximum wave heights.

Following development of the physical theory of underwater explosions, mathematical models were developed by applying inverse methods to experimental time series and simplifying the result to a two-parameter model corresponding to initial conditions on the free-surface elevation ($\eta_0$) representing the maximum surface displacement from the explosive disturbance (Le Méhauté and Wang, 1996). $\eta_c$ corresponds to the maximum depth of the disturbance below equilibrium and $R$ to its radius extent. These parameters physically represent the size of the initial cavity and are functions of explosive yield $E$, water depth $h$, burst depth $z$ and bed characteristics for which calibration is made with empirical data. The initial disturbance can therefore be described analytically by one of a number of candidates for the general profile as Eq. (1-2) (Le Méhauté and Wang, 1996). Note that Eq. (1) is discontinuous at its edge, while Eq. (2) returns back to zero. A schematic diagram illustrates the problem and initial profiles in Figure 1a-c.

$$\eta_0(r) = \begin{cases} \eta_c \left[ 2 \left( \frac{r}{R} \right)^2 - 1 \right], & r \le R \\ 0, & r > R \end{cases} \tag{1}$$

$$\eta_0(r) = \begin{cases} \eta_c \left[ -\frac{1}{3} \left( \frac{r}{R} \right)^4 + \frac{4}{3} \left( \frac{r}{R} \right)^2 - 1 \right], & r \le R\sqrt{3} \\ 0, & r > R\sqrt{3} \end{cases} \tag{2}$$

The relations between parameters $\eta_c$, $R$ and explosive characteristics described here are derived empirically after many series of small and larger scale experimental observations and are well described and reviewed by Le Méhauté and Wang (1996). These relations depend on classifications of water depth $h$ and charge depth below water surface $z$ relative to explosive energy released $E$. In terms of a depth parameter:

$$D = \frac{ch}{\sqrt[3]{E}}, \tag{3}$$

where $c = 406.2$ is an imperial unit conversion constant, three categories are specified when considering wave generation:

$$Classification = \begin{cases} Deep, & D > 14 \\ Intermediate, & 1 < D \le 14 \\ Shallow, & D \le 1 \end{cases} \tag{4}$$

For deep and intermediate cases, the cavity parameters are defined as:

$$\eta_c = aE^{\frac{6}{25}}, \tag{5}$$

$$R = bE^{\frac{3}{10}}, \tag{6}$$

where constants $a$ and $b$ vary as;

$$a = \begin{cases} 0.02913, & 0 < \frac{z}{E^{\frac{3}{10}}} \leq 9.965 \times 10^{-4} \\ 0.01433, & 9.965 \times 10^{-4} < \frac{z}{E^{\frac{3}{10}}} \leq 2.9895 \times 10^{-2} \end{cases} \tag{7}$$

$$b = \begin{cases} 0.03803, & 0 < \frac{z}{E^{\frac{3}{10}}} \leq 9.965 \times 10^{-4} \\ 0.04293, & 9.965 \times 10^{-4} < \frac{z}{E^{\frac{3}{10}}} \leq 2.9895 \times 10^{-2} \end{cases} \tag{8}$$

For shallow cases, it is implicitly assumed that the explosion develops a cavity that extends through the entire water column and exposes the bed, and therefore its radius is larger than the water depth ($R > h$). In this instance, cavity radius is defined as:

$$R = 0.03608 \, E^{\frac{1}{4}} \, . \tag{9}$$

The data calibrating these models include charges ranging from small ($< 500$ lb of TNT or $< 9.5 \times 10^8$ J) to a handful larger ($< 9500$ lb of TNT or $< 1.8 \times 10^{10}$ J) and further include a 23-kT nuclear test (Le Méhauté and Wang, 1996).

### 2.1.1 Volcanic context

For a volcanic case, illustrated in Figure 1d, such explosions would occur on or in an edifice, meaning that the charge depth is equivalent to the water depth at that point ($z = d$), therefore events that are capable of hazardous wave generation fit into the intermediate or shallow depth classes.

An approach first demonstrated by Torsvik et al. (2010) and expanded on by Ulvrová et al. (2014) and Paris and Ulvrová (2019) identifies use of the parabolic initial deformation (Eq. 1) and its compatibility with the shallow class radius definition (Eq. 9) when paired with the intermediate condition of Eqs. (5, 7). This is valid where the released explosion energy is estimated from volcanic crater diameter $C_D$ using the following empirical relationship by Sato and Taniguchi (1997):

$$E = 4.45 \times 10^6 \, C_D^{3.05} \, . \tag{10}$$

This method has recently been used for probabilistic hazard analysis of volcanogenic tsunamis at the Campi Flegrei caldera, Italy (Paris et al., 2019) and at Taal Lake, Philippines (Pakoksung et al., 2021). Some land studies suggest that the size of a volcanic crater cannot be assumed to directly reflect the size of the largest explosion causing it (Valentine and White, 2012), so a further relation from Sato and Taniguchi (1997) relates the ejecta volume $V$ with the released explosion energy:

$$E = 4.055 \times 10^6 \, V^{1.1} \, . \tag{11}$$

### 2.2 Numerical method

To compute simulations of the models described earlier, we use the open source CFD framework, Basilisk (Popinet, 2013). The software is widely used in studies involving multiphase problems from jet dynamics to viscoelastic and surface tension

investigations and includes several free-surface solvers with application to tsunamis, wavefield transformation and other hydrological areas (e.g. Lane et al., 2017; López-Herrera et al., 2019; Berny et al., 2020). The benefits of the Basilisk framework include OpenMP/MPI parallelism capability and an adaptive tree-based Cartesian grid with refinement algorithms that accept user-defined criteria. These allow appropriate resolution of areas of irregular shape or distant from one another within a domain with high scale separation while increasing the overall efficiency and speed of computation. (Popinet, 2011; van Hooft et al., 2018)

### 2.2.1 Multilayer Scheme

The majority of the following work utilised the non-hydrostatic multilayer free-surface solver developed and described by Popinet (2020). A brief outline is given here. The scheme of $n$ layers is a horizontally gridded and vertically discrete approximation of the incompressible Euler equations with a free-surface and gravity. It is described by the system:

$$\partial_t h_k + \nabla \cdot (h\mathbf{u})_k = 0, \tag{12}$$

$$\partial_t (h\mathbf{u})_k + \nabla \cdot (h\mathbf{u}\mathbf{u})_k = -gh_k \nabla \eta - \nabla (h\phi)_k + [\phi \nabla z]_k, \tag{13}$$

$$\partial_t (hw)_k + \nabla \cdot (hw\mathbf{u})_k = -[\phi]_k, \tag{14}$$

$$\nabla \cdot (h\mathbf{u})_k + [w - \mathbf{u} \cdot \nabla z]_k = 0, \tag{15}$$

where, in the $\mathbf{x}$-$z$ reference frame, $k$ is the layer index, $h_k$ layer thickness, $g$ gravitational acceleration, $\mathbf{u}_k$, $w_k$ the horizontal and vertical velocity components, $\phi_k$ the non-hydrostatic pressure, $\eta$ the free-surface height (sum of layer thicknesses and bathymetry height $z_b$), and

$$z_{k+1/2} \equiv z_b + \sum_{l=0}^{k} h_l, \tag{16}$$

the height of layer interfaces.

Between them, the set expresses the evolution of layer thickness (Eq. 12), conservation of momentum (Eq. 13, 14), and conservation of volume/mass (Eq. 15). The framework allows the model to be built modularly, starting from the hydrostatic case where $\phi = 0$ and vertical momentum conservation (Eq. 14) is removed; these effectively become the generalised multilayer Saint-Venant (SV) or stacked shallow water equations. More components are added, for example, vertical remapping, adaptivity, non-hydrostatic and Keller box vertical projections, and a wave breaking method. The latter is implemented by introducing dissipation. This is handled by limiting the maximum vertical velocity by setting

$$w_k^{n+1} \leftarrow \text{sgn}(w_k^{n+1}) \min(|w_k^{n+1}|, b\sqrt{g|H|_\infty}) \tag{17}$$

where $\sqrt{g\,|H|_\infty}$ is the characteristic horizontal velocity scale of the wave and $b$ is a specified breaking parameter smaller than one. $sgn$ and $min$ are sign and minimum functions respectively. The breaking algorithm used throughout the multilayer scheme is described in greater depth in the model's defining paper by Popinet (2020). Lastly, terrain is handled by looking up a pre-processed k-dimensional tree indexed database of heights and the model is able to discriminate and resolve areas of wetting and drying.

The scheme has been tested against numerous benchmark cases, including standing waves, sinusoidal wave propagation over a bar, the Tohoku tsunami of 2011 and its dispersive features, viscous hydraulic jumps, and breaking Stokes waves (Popinet, 2020). Source code of these examples can be found at http://basilisk.fr/src/layered/nh.h#usage. Able to provide accurate solutions for surface gravity waves across a wide range of scales, the multilayer scheme is ideal for resolving the evolution of an explosive source initial condition, resulting wave propagation and shore run-up (Popinet, 2020).

### 2.2.2 Other Schemes

Accompanying models used for validation and comparative purposes include a Volume-of-Fluid method (N-S/VOF) in the same framework which solves the two-phase Navier-Stokes equations for interfacial flows, including variable density and surface tension (Popinet, 2009, 2018), a solver for the shallow water or Saint Venant equations (SV), and finally another for the Serre-Green-Naghdi equations (SGN), a Boussinesq higher order approximation for non-linear and weakly dispersive flows (Popinet, 2015). Their inclusion is to support and inform evaluation of the multilayer scheme against well known and commonly used methods.

In this work's context, the main discriminations between the schemes used are both the hydrostatic assumptions involved and their resolution of vertical gradients such as the velocity profile: in the hydrostatic SV solver, a constant velocity profile is assumed; the non-hydrostatic SGN equations represent a Boussinesq-type analytical approximation of the vertical structure; the multilayer method resolves the vertical to $n$ layers with capability to include non-hydrostatic terms; the N-S/VOF scheme fully resolves the vertical as an additional dimension. The addition of having to fully solve a 2D slice through the vertical versus 1D means the N-S/VOF has a larger domain to compute for the same region when considering water waves.

Only recently have non-hydrostatic multilayer methods advanced towards a point where their stability and efficiency has approached or challenged Boussinesq-type schemes. These are typically used for applications such as landslides which can generate highly dispersive (high $kh$) initial waves and significant vertical accelerations. A preferred current method is often to initialise the wave generating source, usually a short duration compared to the total tsunami simulation length, using either a direct numerical method (e.g. Abadie et al. (2012)) or a NH multilayer scheme (e.g. Grilli et al. (2012, 2019); Schambach et al. (2021); López-Venegas et al. (2015)), and then to interpolate or otherwise insert the results into an initialisation of a Boussinesq-type scheme to further propagate the waves across a larger domain.

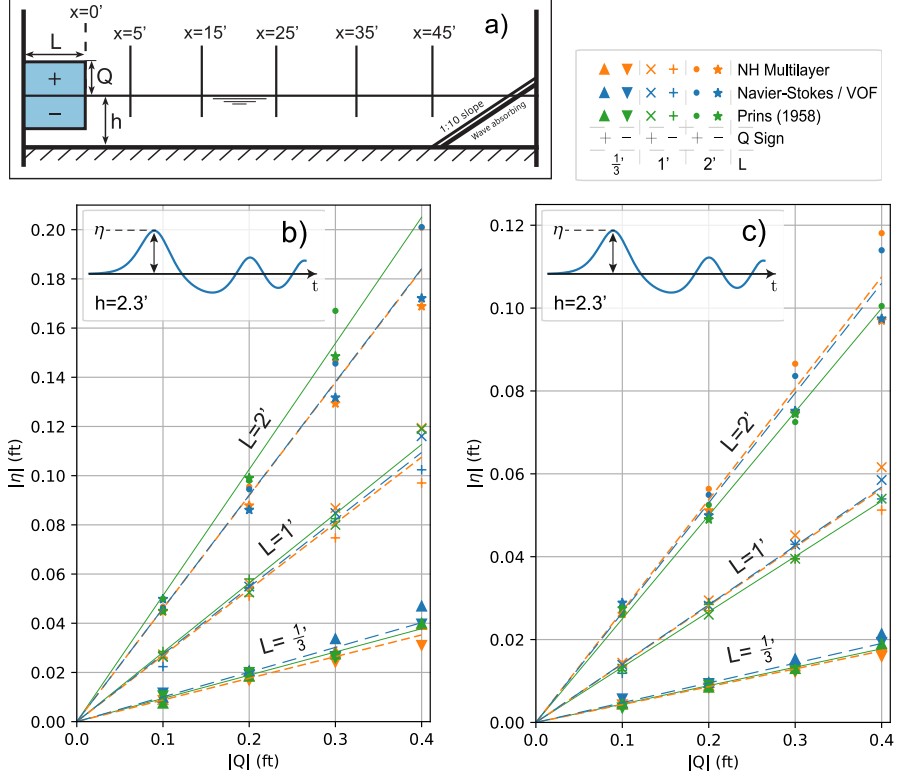

**Figure 2.** (a) Schematic diagram of Prins (1958) experimental set up, with gauge locations and the initial disturbance. (b) $\eta$-$Q$ relation for first arrival crest or trough amplitude at $x = 5$ feet. (c) $\eta$-$Q$ relation for first arrival crest or trough amplitude at $x = 25$ feet. Regression lines through origin are plotted for each $L$-model combination.

## 3   Laboratory-scale validation

To determine suitability of the numerical method for use in modelling a subaqueous disturbance, validation against a suitable case study is required. Prins (1958) conducted a flume experiment investigating surface waves produced from an instantaneously raised or depressed column of water. This case is replicated using the multilayer scheme and, additionally, the N-S/VOF, Saint-Venant and SGN schemes for full numerical comparison. Unless otherwise specified, all length units in this section only are in imperial standard for consistency and ease of comparison with the Prins (1958) dataset.

The experimental set up involved a column of water at rest of extent $L$, height from equilibrium $Q$ and in a one-foot wide flume of water depth $h$; these parameters were varied by steps: $L = 2, 1, 1/3$ ft; $Q = \pm 0.1, \pm 0.2, \pm 0.3$ ft; $h = 2.3, 0.5, 0.35, 0.2$ ft. Figure 2a illustrates a schematic diagram of the setup. The height-adjusted column was held by a difference in air pressure and released by means of a sliding door opened by hand in under 0.03 s. The resulting waves are measured at five gauges positioned every 10 ft starting at 5 ft. A sloped 'wave absorber' dampened reflections from the flume end.

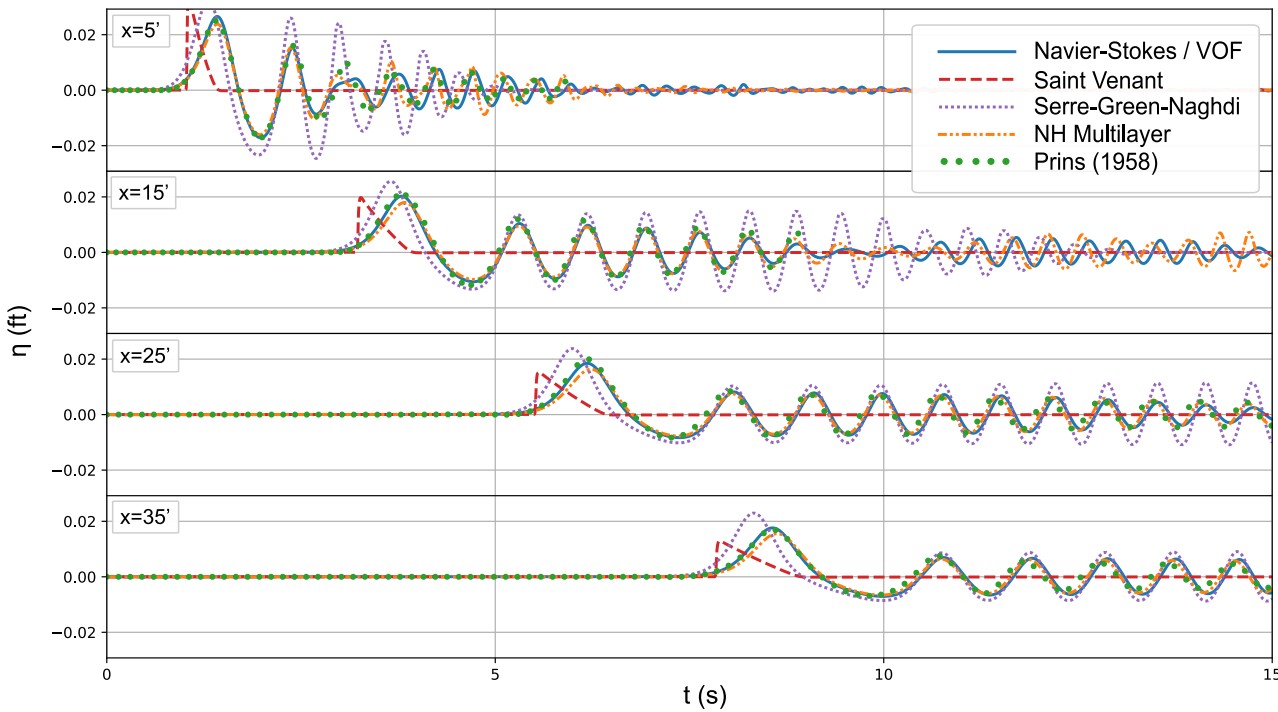

**Figure 3.** Model comparison of time series of waves produced by a disturbance of $Q = 0.3$, $L = \frac{1}{3}$ when $h = 0.5$.

The numerical models to replicate this are relatively simple. For the multilayer, SGN and SV schemes, a 1D domain of
length 100 ft is initialised with the origin translated by $L$, the positive part of the domain with initial water depth $\eta_0 = h$, and the negative with $\eta_0 = h + Q$. The displaced column is released instantaneously. While a reflective boundary condition is applied to both ends, extending the positive domain far beyond the experimental case to prevent reflections has negligible computational cost. In the multilayer simulations, 20 regularly spaced layers were used and the breaking parameter, following guidance and methods used in Popinet (2020), is set to $b = 0.38$. The N-S/VOF scheme's geometry is built similarly but in two-
dimensional vertical (2DV), such that the origin is translated by $(-L, -h)$. Material properties (i.e. density, dynamic viscosity, surface tension coefficient) of the two phases are also specified, as is gravitational acceleration. For all presented cases, unless otherwise specified, the maximum level in the grid is 14, which, for a domain of length 100 ft, gives a maximum horizontal mesh resolution of $6.1 \times 10^{-3}$ ft.

### 3.1    Model comparison

Figures 2b-c and 3 present direct comparisons between the numerical models and the published experimental data. The N-S/VOF and multilayer approaches each conform well when varying $L$ and $Q$ at $h = 2.3$ ft and generally confirm the experimental findings for initial wave amplitude throughout the flume range. Some experimental data points at high $Q$ are missing;

however, the model output highlights a slight divergence in first arrival amplitudes between +ve (higher) and -ve (lower) disturbances at larger $Q$.

Figure 3 plots the full time series for a mid-range parameter run and includes solutions from the non-linear shallow water (SV) and Boussinesq-type (SGN) schemes. While the SV solution exhibits a good approximation of the phase velocity of the initial wave (4.39 ft s$^{-1}$), it is, unsurprisingly, unable to resolve any of the trailing wave field. The SGN scheme resolves this element at far-field well; however, it often overestimates initial wave amplitude and the overall dispersivity. The multilayer scheme is shown to be very accurate compared to the N-S/VOF result and the experimental trace and reinforces the good fit found in Figure 2. This suggests that the scheme faithfully replicates the process of a collapsing water column or infilling of a uniform depth and the associated wave generation, along with reflections from behind the initial disturbance.

The generation process is shown in Figure 4 which illustrates how the four models resolve an initial disturbance and reveals significant variations in handling the vertically critical jump in the first half-second. The SV source quickly develops into the classic steep-fronted crest often seen in dam-break problems where the amplitude is proportional to the initial disturbance height. The higher-order SGN scheme is able to resolve frequency dispersion and therefore handles the source development better; however, there are clear exaggerations such as at 0.05 s near $x = -L$ where, intriguingly, the water height temporarily increases above $Q$. Qualitative comparison between the development of both the N-S and multilayer solutions show a high consistency across all stages involving the collapse of initial disturbance and breaking of the generated wave. Notable variation includes the varying maximum amplitude of the resultant wave's intermediate stage and any bubbles/droplets produced, demonstrating the restriction of a single value for function on the 1D multilayer scheme not present on a 2D multiphase VOF solver, meaning that there are many neglected phenomena in the multilayer scheme such as bubbles and plunging breaks that, while this may be negligible in this lab-scale experiment, could be more significant at larger scale. Also, note graphical artefacts on free-surface height at the leading gradient in the N-S solution which occur at coarser regions of the adaptive grid. Despite these minor variations, it is clear that the multilayer model has greater validity in application to this case than either of the single layer models and is remarkably consistent with the physical experiment as well as the directly simulated approach.

Finally, Table 1 presents performance metrics for the numerical schemes across three maximum refinement levels for an example run case. All were performed with OpenMP parallelism on eight CPU cores. The multilayer scheme fits between the SV and SGN methods in terms of wall time processing and offers a vast improvement in computational efficiency compared with the N-S/VOF solver considering result similarity. It is also faster than the dispersive SGN method, primarily due to the computation time required to solve the higher order approximation.

## 3.2 Wavefield classification

All simulated cases are plotted in Figure 5 by the size of initial disturbance relative to water depth using a nonlinearity parameter $\frac{|Q|}{h}$ against a dispersivity parameter $k*h$ where $k*=\frac{2\pi}{2L}$. Six additional simulations were run outside of the original experimental parameter space to expand the model dataset reach. As done by Prins (1958), the $+Q$ runs are categorised into groups with similar wavefield characteristics starting with strong oscillatory properties (blue) where $k*h > 10\frac{|Q|}{h}$, tending through increasingly solitary wave properties (purple and green) once $k*h \leq 10\frac{|Q|}{h}$ until a succession of diminishing ampli-

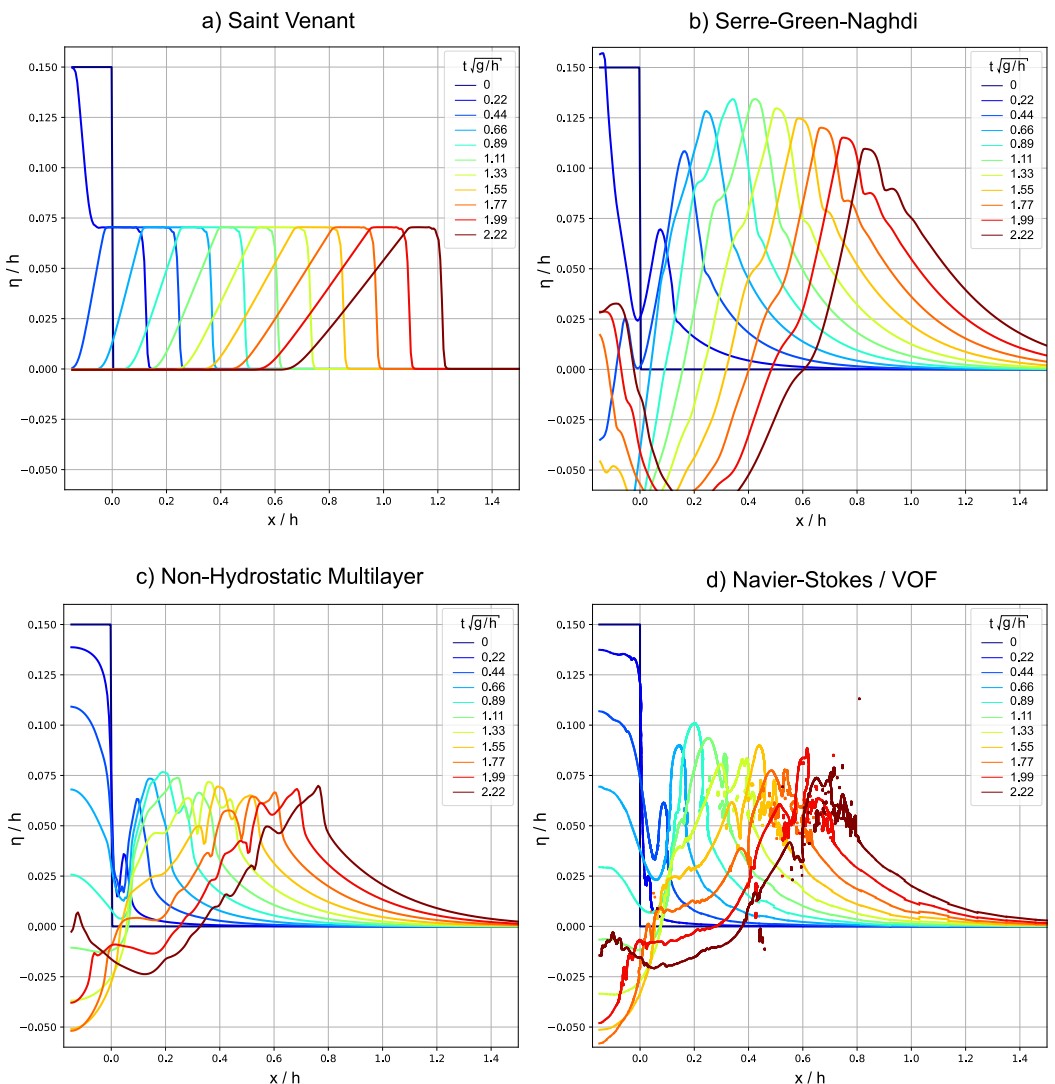

**Figure 4.** Comparison of numerical solutions for the disturbance collapse from initialisation to $t = 0.5$ seconds. Distances are normalised with water depth $h$ and are vertically exaggerated.

tude solitary waves result where $k*h < \frac{|Q|}{h}$ (orange). Beyond this region, the initially generated bore survives far enough down the numerical flume before it would likely separate (black). For the $-Q$ domain, all resultant wave fields were similar except for the length of initial trough relative to the following periodic wave group. This ratio initially remains approximately unity in the same region as the $+Q$ oscillatory character group and grows larger towards higher $\frac{|Q|}{h}$ and lower $k*h$.

Results of the $+Q$ disturbance (Figure 5a) corroborate the experimental wave field descriptions of Prins (1958), including the transition of an oscillatory field through to solitary initial waves. Bore formations in the first stages were also observed during the experiment, those of which last a considerable length of the flume match model results. Accounting for tolerance

**Table 1.** Table of numerical model performance for run $h = 2.3'$, $L = 0.2'$, $Q = 0.3'$. Runtime in seconds and speed in $points \times timesteps \times layers/sec/cores$.

| Scheme | Level | Runtime | # Timesteps | Speed |
|---|---|---|---|---|
| Saint Venant | 13 | 93 | 44049 | 477513 |
| | 14 | 231 | 88167 | 478551 |
| | 15 | 1022 | 176414 | 496146 |
| Serre-Green-Naghdi | 13 | 215 | 44077 | 209928 |
| | 14 | 825 | 88779 | 220389 |
| | 15 | 2444 | 179392 | 300650 |
| Multilayer (10 layers) | 13 | 125 | 4407 | 358552 |
| | 14 | 477 | 7017 | 301021 |
| | 15 | 1763 | 11392 | 264652 |
| Multilayer (20 layers) | 13 | 379 | 5183 | 279903 |
| | 14 | 957 | 11874 | 507919 |
| | 15 | 2935 | 16123 | 450039 |
| Navier-Stokes / VOF | 13 | 4174 | 4276 | 15834 |
| | 14 | 22751 | 12686 | 14840 |
| | 15 | 126279 | 27859 | 13705 |

in qualitative descriptions, groups match closely and the additional results beyond experimental scope further confirm these definitions in the studied range. Such an analysis of the $-Q$ part was not attempted in the original research; however, effort in this area can be made with the numerical results (Figure 5b). The length ratio of initial trough to the following oscillatory waves increases with higher $Q/h$ and lower $k*h$. This matches the trend towards solitary characteristics with $+Q$. Intriguingly, this pattern holds regardless of the length ratio that defines the initial disturbance (i.e. $Q/L$).

These results bear similarity to those found by Hammack and Segur (1974, 1978a, b) in experiments involving a piston producing vertical bottom motions described by Hammack (1973) and modelling using the Kortewig-de Vries equation, also with an initialised rectangular wave source. Notable similitudes include the generation of potentially multiple solitons of decreasing amplitude for shallow water positive initialisations (as seen in orange region of Figure 5a), followed by a train of dispersive oscillatory waves and that no solitons are generated from negative vertical motions, instead producing a wave train of the type illustrated in Figure 5b of an initial 'triangular' wave of greater speed than the trailing modulated oscillatory waves.

In suitable replication of the experimental findings, the present numerical scheme is seen to be fit for generating accurate waves from initialised disturbances and modelling their near-field propagation across a significant regime range where non-

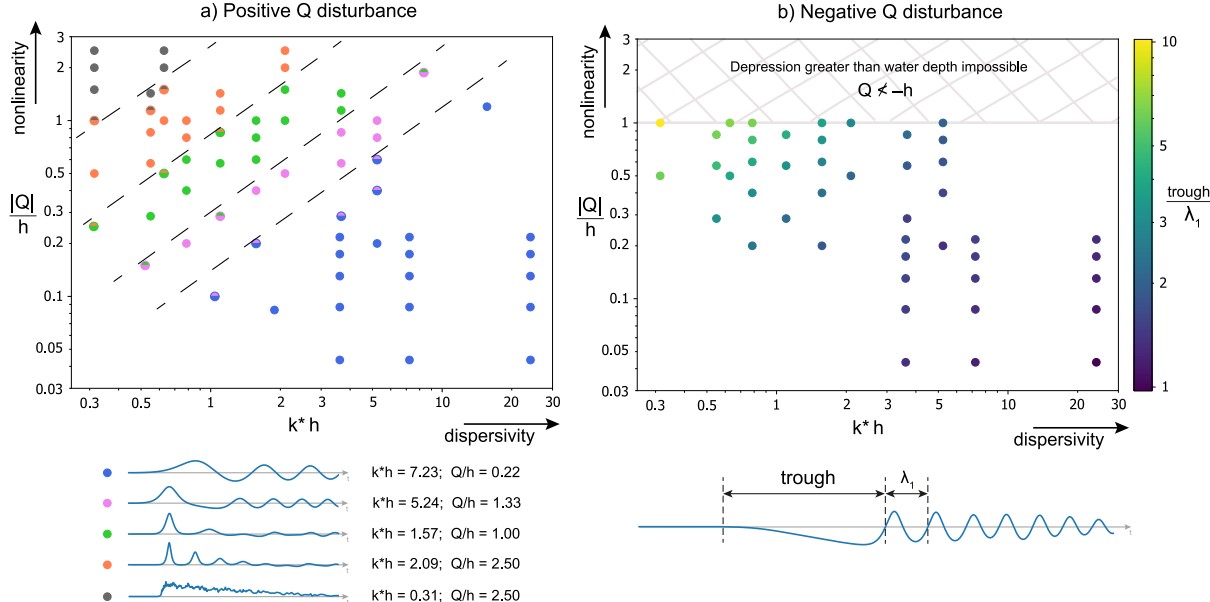

**Figure 5.** Plots of numerical runs by dispersivity parameter against nonlinearity parameter. All data from time series at $x = 35$ feet. (a) Classification of $+Q$ disturbances by their generated wave field. Legend underneath is illustrated by plots of example runs. (b) $-Q$ disturbances coloured by the ratio of initial trough length divided by the following wavelength.

linear and dispersive effects may be prevalent. The method also demonstrates suitability for further investigations either beyond or in complement, such as to widen parameter spaces with relatively low computational expense.

## 4    Field-scale validation

The next stage is to assess use of the underwater explosion models of Sect. 2.1 within the numerical scheme. To do this, we utilise datasets from the Mono Lake test series in 1965, conducted by the Waterways Experiment Station, and documented by Walter (1966); Wallace and Baird (1968); Whalin et al. (1970); Pinkston et al. (1970). This was one of the largest chemical explosive test series designed to investigate subsequent water wave generation and shore effects. A series of ten approximately 9,250 lb (~4,196 kg) spherical TNT charges were detonated off the south shore of Mono Lake, California. The test area is illustrated in Figure 6a, and this also shows the location of wave gauges arranged in four radials directed away from ground zero (GZ), along with contoured terrain. GZ was located at a site of approximate water depth $h = 39$ m.

The numerical model was built with the multilayer scheme, including a digital elevation model of the Mono Lake bathymetry by Raumann et al. (2002), where the lake water level was set at elevation above mean sea level 1945.7 m. This left approximately 2 m vertically of the bathymetric model dry to act as the shore surrounding the lake. Numerical gauges were defined

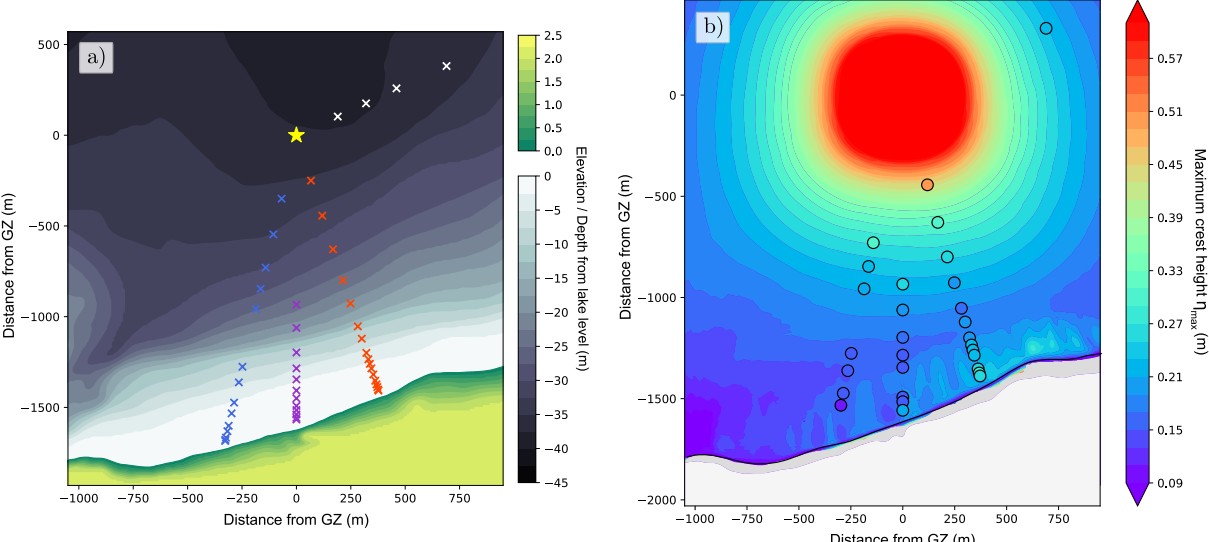

**Figure 6.** (a) Setting of test series on the south of Mono lake with bathymetry and near shore terrain elevation. Yellow star marks ground zero (GZ), crosses mark gauge locations. Gauge radials are coloured by; red (one), purple (two), blue (three), white (four). (b) Numerical results of maximum crest amplitude for Shot 3 overlaid by observed maximum crest amplitudes at gauge locations.

**Table 2.** Numerical values used in Mono Lake models.

| Shot | Energy $E$ ($\times 10^{10}$ J) | Charge Depth $z$ (m) | $\frac{z}{E^{\frac{3}{10}}}$ | $a$ | $b$ | $\eta_c$ (m) | $R$ (m) |
|---:|:---:|:---:|:---:|:---:|:---:|:---:|:---:|
| 3 | 1.755 | 0.427 | $3.61 \times 10^{-4}$ | 0.02913 | 0.03803 | 8.36 | 44.90 |
| (75% Yield) 3 | 1.317 | 0.427 | $3.93 \times 10^{-4}$ | 0.02913 | 0.03803 | 7.67 | 41.18 |
| 9 | 1.739 | 6.401 | $5.42 \times 10^{-3}$ | 0.01433 | 0.04293 | 4.12 | 50.83 |
| (75% Yield) 9 | 1.304 | 6.401 | $5.91 \times 10^{-3}$ | 0.01433 | 0.04293 | 3.85 | 46.62 |

at distances from GZ described in the experiment. The maximum horizontal resolution across the domain was 1.46 m and simulations used 5 layers.

Data from shots 3 and 9, individual explosive tests within the experimental series, are chosen to test the numerical method against. These experiments resulted in the highest quality data and were detonated at different depths, $z_3 = 0.427$ m and $z_9 = 6.401$ m. Following a similar method used by Paris et al. (2019), the initialised disturbance takes the form of Eq. (2) and, for these charges, $D = 6.1$ therefore the depth classification (Eq. 4) is intermediate. The resultant values using Eqs. (4-7) are given in Table 2. Considering doubts regarding the charge magnitudes raised in the report of Walter (1966), additional lower yield simulations were added.

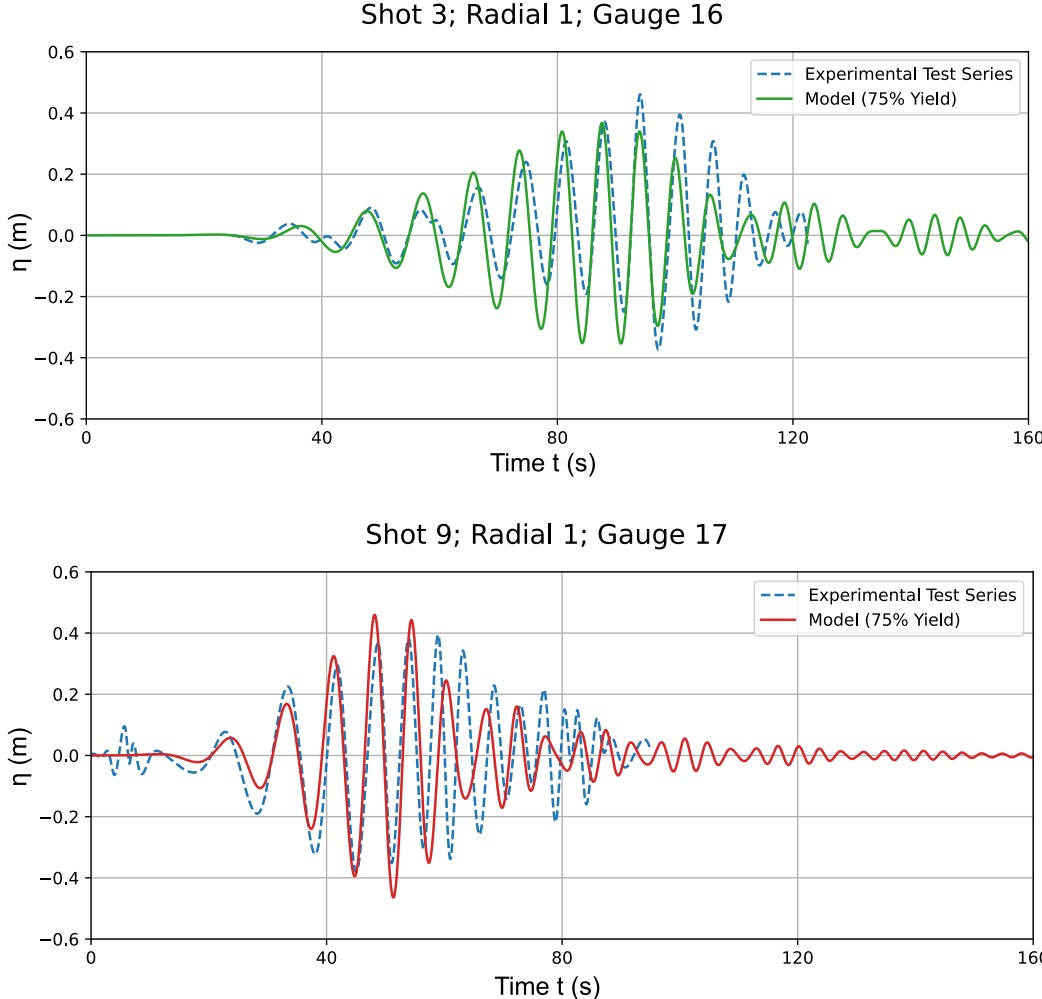

**Figure 7.** Time series comparison at closest gauge to GZ for both considered shots.

## 4.1 Generated wavefield

Figure 6b illustrates the generated field of maximum crest amplitudes from the numerical simulation paired with data tran-
scribed from the experimental gauge records for Shot 3. This, and all other included simulations, are presented quantitatively
in Figure 7 to show maximum crest amplitudes for all gauges. The maximum amplitude of the initial envelope decreases with
radial distance from GZ as expected, matching experimental observations. Shoaling is most pronounced at the closest incident
shoreline, in the region 500 m east of Radial 1, and becomes far less significant with distance as seen on the western shore of
the region. Maximum crest amplitudes at gauge locations follow similar patterns in all runs; however, the higher energy yield
simulations produce greater amplitudes (additional 0.11–0.05 m) throughout and experience more significant shoaling in all

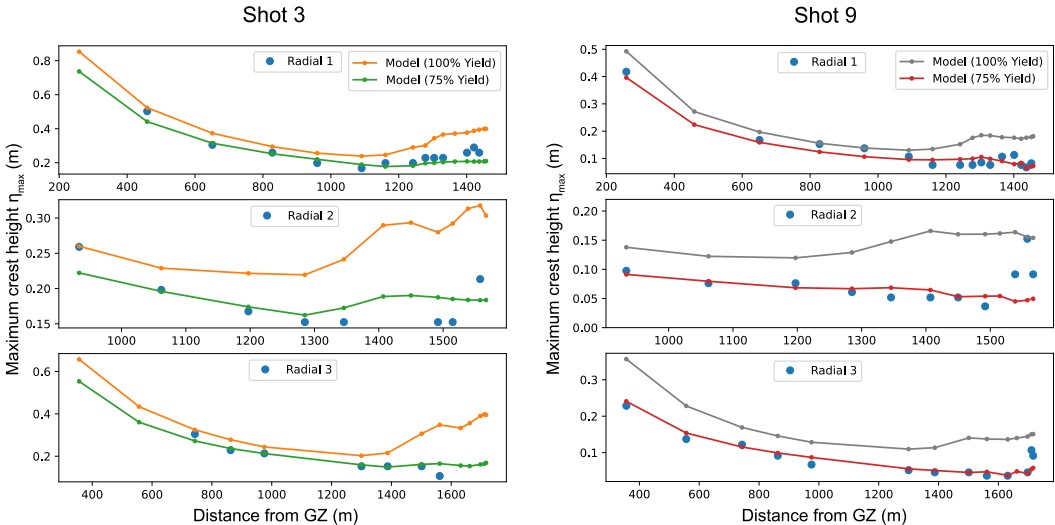

**Figure 8.** Maximum crest amplitudes at gauge locations for full yield model, fitted model, and experimental tests.

shallow areas. The lower yield simulations are a closer match to experimental observations for both shots, especially in shallow zones; however, the experimental data has noticeably greater variation at shore.

The experimental gauge time series at locations closest to GZ are plotted alongside the lower yield model traces in Figure 8 and are useful to compare the phase arrival times and the initial development of the wave train. For both simulations, the first
arriving phases match the experimental record very well, with the exception of minor noise immediately following detonation in Shot 9 which is likely to be shock or debris related. The time of maximum envelope amplitude also conforms well, where the differences are 9 and 7 seconds for Shots 3 and 9 respectively. The latter part of the initial wave group maintains higher amplitudes in the experimental trace for both records, whereas the envelope decay is sooner in the numerical model. Shot 3 also seems to exhibit a positive amplitude shift in the early part of the experimental envelope. These could be genuine
underestimation of wave amplitudes towards the end of the initial group which could be due to variations in dissipation by breaking waves of the initially generated energy.

In terms of parameters pertaining to dispersion and nonlinearity, $kh$ and $ka$, waves in the initial group near the source at the nearest gauges on each radial were in the ranges $1.34 < kh < 3.56$ and $3 \times 10^{-3} < ka < 1.181$ and across the gauges beside the shore were in the ranges $0.105 < kh < 0.235$ and $3.6 \times 10^{-3} < ka < 1.2 \times 10^{-2}$. As would be expected, moderately nonlinear
waves are generated and $kh$ decreases as the waves approach the shore and become shallow. Wave steepness $ka$ immediately decreases when propagating away from source across near-flat bathymetry, but then increases on average as water depth $h$ decreases towards shore.

## 4.2 Model implications

While the models used fit the experimental results and trends well overall, it is significant that the lower yield data is a much better fit. In reports following the test series (e.g. Whalin et al. (1970)) it is noted that, except for two charges, all shots delivered below average expected maximum crest amplitudes as predicted by earlier experimentation, particularly the deep water shots. Further similar deep tests at Mono Lake in the following year, which were conducted for other investigations, delivered much greater amplitude waves in line with expectations, leading to the suggestion by Wallace and Baird (1968) that, beyond scaling effects or measurement issues, the charges used in the series may have been faulty and delivered a lower yield. This is supported by the numerical data as, when energy is reduced, the maximum crest amplitudes can be accurately predicted in addition to the resultant early wave group and individual phases. Moreover, further energy dissipation not accounted for in the physical model may be responsible for the greater fit of the reduced yield simulation, such as losses from a higher amount of dissipation from breaking of the initial waves from the explosion or some of the energy transferred to the nearby bed as elastic deformation. While the initialisation model is calibrated to charge depths relative to water depth, bed characteristics were not strongly considered.

Many data from the experimental series were unreported or discarded following the series due to various problems, including excess noise generated from the explosion itself or wind-driven waves and are thus missing from comparisons in this work. Instrumental issues meant gauges near the shore were often unable to be acceptably calibrated for amplitude. Many experienced +ve noise due to wave breaking and bores, responsible for some spikes in measurements; however, these were kept for reliable arrival times and periods. (Wallace and Baird, 1968)

Despite the experimental challenges, the multilayer scheme is shown to be good at resolving the initialised disturbance into a wave field very consistent with that generated by an intermediate depth submarine explosion. With the capability to accurately propagate such a source in the relatively deep ($kh \approx 3$) near-field through to the shore, it demonstrates suitability to model such events at these spatial and time scales. However, this test did not contain any shallow depth underwater explosion tests, for which there are no case examples at this scale.

## 5 Taupō Scenario

We now apply the multilayer model presented earlier to a hypothetical sublacustrine eruption to demonstrate a case-use example for a potential hazard study. Lake Taupō is New Zealand's largest freshwater lake of area approximately 616 km$^2$. It lies in the southern section of the Taupō Volcanic Zone and conceals most of Taupō volcano and the caldera of Earth's youngest supereruption (Oruanui) at c. 25.5 ka, ejecting $> 1100$ km$^3$ of pyroclastic material (Davy and Caldwell, 1998; Wilson, 2001; Vandergoes et al., 2013; Allan, 2013). Taupō volcano and its post-supereruption magmatic system have been extensively studied because of how recently the event occurred, along with the volcano's relatively high activity. As a result, the eruptive history of Taupō is well constrained through radiocarbon dating methods, stratigraphy, and composition analysis (Wilson, 1993; Wilson et al., 2009; Barker et al., 2015).

The Oruanui event itself was the evacuation of a rapidly formed (<3000 years) and well mixed high-silica rhyolite melt dominant body (Wilson et al., 2006; Allan, 2013; Wilson and Charlier, 2009; Allan et al., 2013) which left the remaining mush source heavily modified by intrusion of hot mafic magmas (Barker et al., 2014, 2015). Following a notably brief 5 kyr quiescence, three further, much smaller (0.004 to 0.05 km$^3$ dense rock equivalent, DRE) dacitic events occurred from ~20.5-17 ka which preceded another 5 kyr gap before an additional 25 rhyolitc eruptions from ~12-1.8 ka, which are all remarkable

in their high variation in eruption volumes (0.015 to 35 km$^3$ DRE) and concentration of vent locations, shown in Figure 9 (Gelman et al., 2013; Wilson, 1993; Sutton et al., 2000; Barker et al., 2015). Compositions of these eruptions reveals a thermal reset of the magmatic system, with more crustal material incorporated into the younger magmas in the series (Barker et al., 2014, 2015). The largest of these, the Taupō plinian eruption in ~232 CE, was one of the most powerful events globally in the past 5000 years (Wilson and Walker, 1985; Houghton et al., 2010) and resulted in the further collapse of the caldera beneath

the lake and, afterwards, the formation of the Horomatangi Reefs (Davy and Caldwell, 1998).

After the last events, the lake underwent refilling and eventually breakout down a main outlet, the Waikato River (Manville et al., 1999; Manville, 2002). In modern times, the lake has impeded monitoring and investigation of the volcanic system state, but this has improved with developments in seismology, geodesy and other geophysical techniques, including identifying numerous seismic swarms and ground deformation in 1983, 1997, 2008 and 2019 (Barker et al., 2020; Illsley-Kemp et al., 2021).

This, accompanied with a good understanding of the varying eruptive history informs the current state of Taupō. However, the lack of relationship between repose and a potential eruption size make it challenging to infer future activity. Therefore, any attempt at scenario-based modelling involving Taupō volcano should consider plausible cases across the full span of statistical likelihood. Here we model a single eruption from Taupo to demonstrate the multilayer model, but highlight that future work will be required to provide an assessment of tsunami hazards from this volcano.

## 5.1  Model set-up

Figure 9 illustrates the lake bathymetry and surrounding terrain accompanied by geological features and settlement locations. The lake is fed by multiple rivers and notably from Tongariro hydroelectric power station via the Tokaanu Tailrace Canal. The sole outflow is the Waikato River, controlled by gates at the largest settlement on the lake, Taupō, which leads to numerous further hydroelectric dams downstream. Surrounded by abundant geothermal resources, strong trout fishing and agricultural

industries, and the area also boasts plentiful tourism opportunities, hosting over one million tourists each year.

In building a model representing an example tsunamigenic explosive eruption in Lake Taupō, it was required to build a terrain dataset by combining a bathymetric model of the lake (Rowe et al., 2002) with an elevation model generated using LiDAR datasets from the Waikato Regional Council which cover the entire foreshore. The limiting resolution of the resultant digital terrain model is that of the bathymetric model (10 m), therefore the simulations were performed at a grid refinement

level of 11 resulting in a horizontal resolution of 16 m.

The eruption site chosen is within the region where most Holocene vents are located and in an active geothermal field (Bibby et al., 1995; De Ronde et al., 2002). Considerations made for the size of eruption involve selecting a size which is plausible with respect to the eruptive history. A single, gigantic blast corresponding to an instantaneous release of material equivalent to

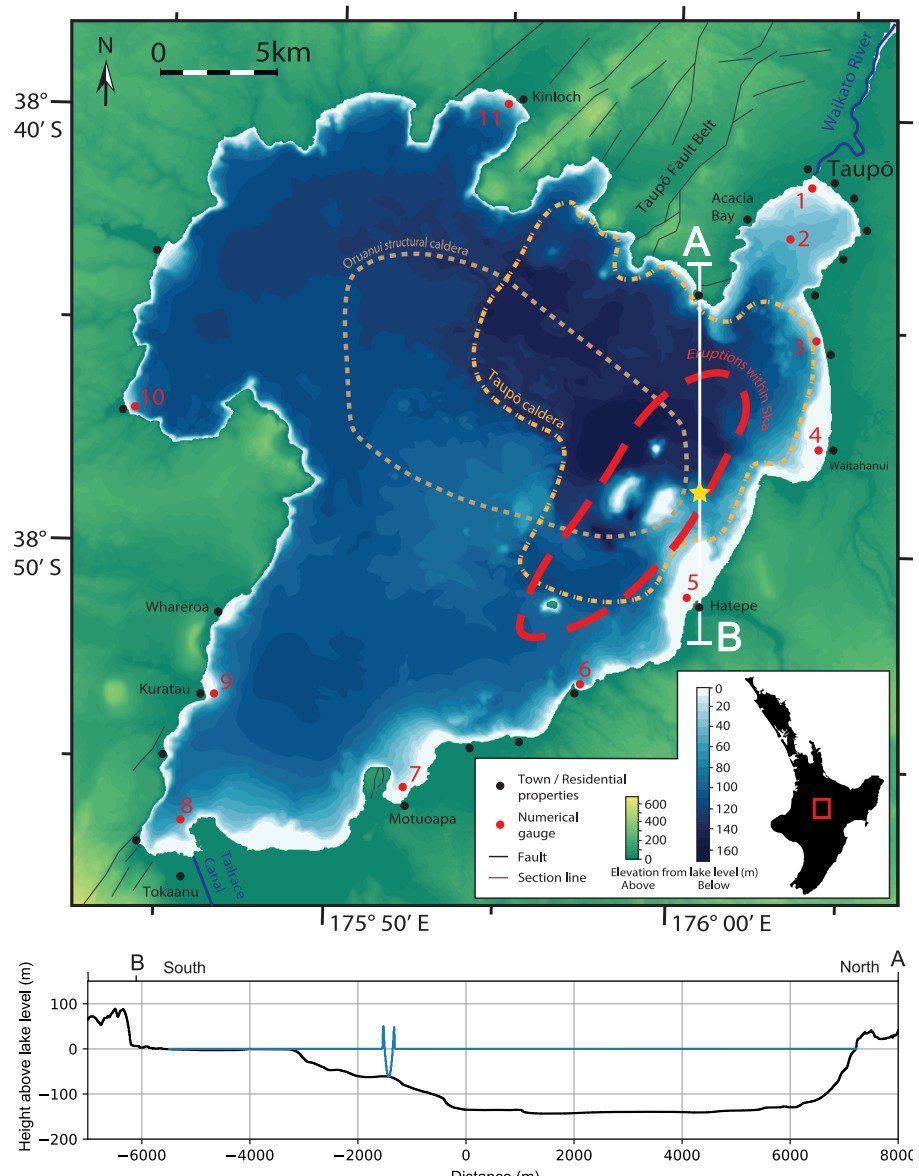

**Figure 9.** Lake Taupō setting with lake bathymetry, surrounding terrain, marked fault zones and built-up areas. Also highlighted are the caldera zones of the Hatepe / Taupō eruption and the Oruanui supereruption (Leonard et al., 2010). The red dashed area encloses area of activity within 5 ka from Wilson (1993); Barker et al. (2015). The star indicates simulated eruption location at -38.813° S 176.016° E. Below is a cross-sectional profile of high run initial condition with terrain elevation in a N-S direction in line with source location.

a larger eruption is not realistic as eruptions of such size typically endure for hours or days (Pyle, 2015). Therefore, for this example, a simulation is run with an ejecta volume corresponding to a plausible mass eruption rate (MER) of $1.2 \times 10^7$ kg s$^{-1}$ which is typical of a volcanic explosivity index (VEI) 4 intensity eruption (Barker et al., 2019). Over the simulation time

**Table 3.** Numerical values used in Lake Taupō models.

| MER $(kg\ s^{-1})$ | Ejecta volume $V$ $(km^3)$ | Energy $E$ $(J)$ | Depth $h$ $(m)$ | $\eta_c$ $(m)$ | $R$ $(m)$ |
|---|---|---|---|---|---|
| $1.2 \times 10^7$ | 0.004 | $7.41 \times 10^{13}$ | 65 | 63 | 105.90 |

(1000 s), the estimated ejecta volume $V$ is 0.004 km$^3$ and, using Eqs. (9-10), the equivalent explosive energy can be obtained. At this depth ($h = 65$ m), all events of $V > 0.0012$ km$^3$ yield $D < 1$, falling into the shallow depth class. Following the rest of the method in Sect. 2.1.2, model parameters for both models are detailed in Table 3. A further simulation of the scenario was performed with the Saint Venant scheme with identical terrain and model geometry for comparison.

## 5.2 Generated wavefield and shoreline impacts

Figure 10 describes the travel times and maximum crest amplitudes. As demonstrated earlier, a succession of waves radially propagate outwards from the centre of the explosion. The initial phase velocity typically starts from $\sim$40 ms$^{-1}$ until they heavily interact with the nearby Horomatangi Reefs. Due to the limited area of the lake, the entire shoreline experiences wave phenomena from these sources within 15 minutes. Long wavelength oscillations (up to seiche) linger throughout the lake beyond simulation length (17 min).

The initial wave at all locations is a crest, reflecting the positive amplitude lip of the initial condition. Earliest arrivals at shore occur about the closest point east of the source at 4 min, after which arrival time generally scales with radial distance excepting for areas with extended shallow zones. The highest wave heights incident to the shore are, unsurprisingly, located nearest the event on neighbouring eastern and northern shores where crest amplitudes reach over 0.11 m. The lowest are found in the further area of the south-west besides Gauge 8 (Tokaanu) and in sheltered parts away from direct paths such as by Gauges 10 and 11 (Kinlock). Taupō township is relatively sheltered compared to the surrounding shoreline due to shielding from the lake morphology.

Figure 11 presents numerical gauge time series for both runs. Throughout the domain, the high ejecta volume run returns significantly higher maximum crest amplitudes than the low run. Wave periods vary from $\sim$40 s early in group to $\sim$15 s towards its end, 10 min later. The first arrival is generally the longest period wave but rarely the greatest amplitude at the gauge locations. Gauge 4 and 5, positioned in shallow zones near the eruption, initially record initial waves of amplitude 0.017 m and 0.04 m before waves of 0.02 m and 0.11 m respectively. Wave generated by this source are initially highly dispersive within 5 km, with waves through the group in the ranges $2.17 < kh < 5.57$ and $4.3 \times 10^{-4} < ka < 8.5 \times 10^{-3}$, and at approaching nearby shores (e.g. most numerical gauges shown in Figure 9) they become more non-linear, exhibiting values in the ranges $0.03 < kh < 1.35$ and $1.8 \times 10^{-4} < ka < 5.7 \times 10^{-3}$.

The resultant waveforms produced from this source are similar to those in Sect. 3 and indicate that the multilayer scheme is suitable for modelling their propagation. The combinations and transformations between highly dispersive, non-linear and shallow waves across the whole domain require treatment that would likely lack validity if only modelled using the shallow

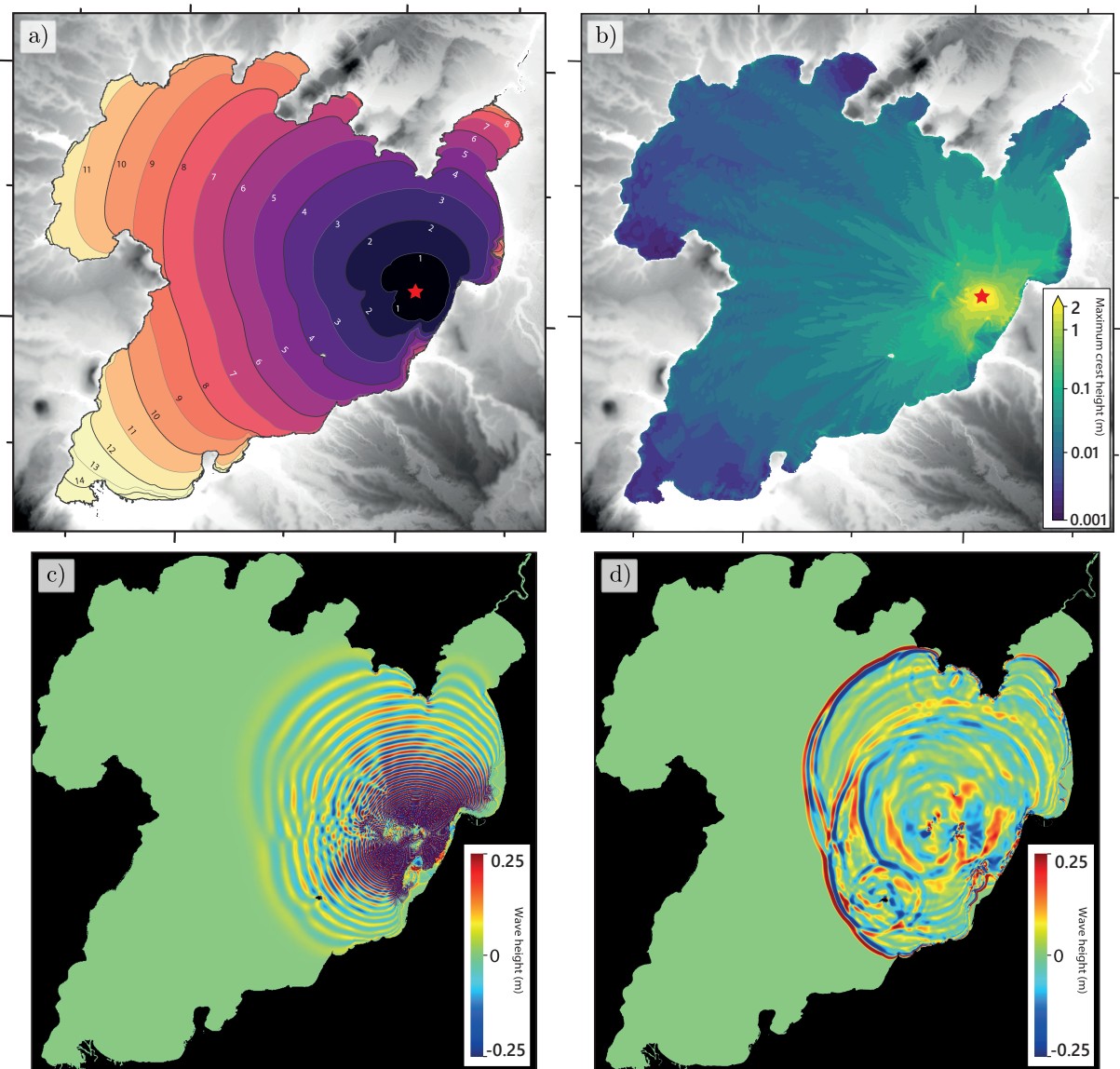

**Figure 10.** Multilayer simulation of potential volcanic explosion in Lake Taupō. (a) First arrival travel times in minutes. (b) Field of maximum crest amplitudes. (c) Snapshot of wave amplitudes at $t = 4.5$ minutes. (d) Equivalent of c) but using the Saint Venant scheme.

water equations. A snapshot of this comparison is illustrated in Figure 10c-d, and clearly shows the heavy dispersion modelled in the multilayer scheme which is missing in the Saint Venant method. The benefit of non-hydrostatic pressure terms and numerical wave breaking therefore adds robustness to the near-shore solution for this type of source. Notably, the explosion model could prescribe an initial condition that intersects the bathymetry in a higher ejecta volume case, which would result in additional mass added via the volume of lip surrounding the cavity. It would be expected that a higher energy explosion in

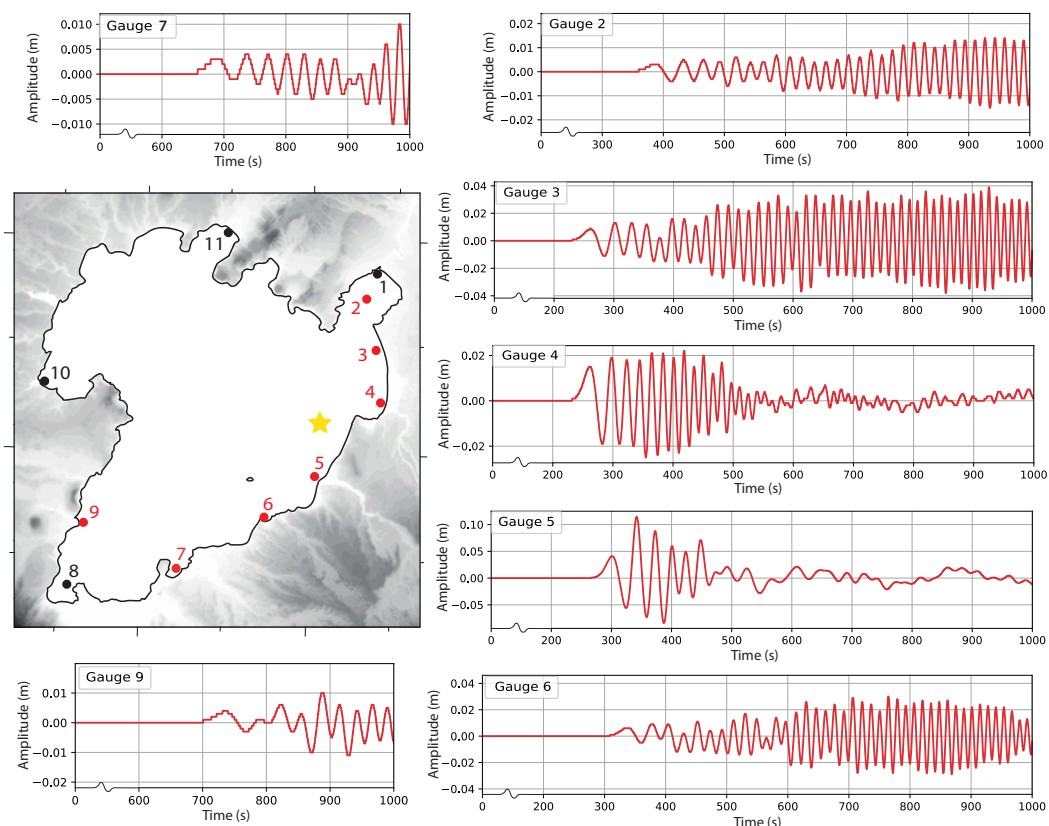

**Figure 11.** Time series for numerical gauges with gauge locations highlighted in red on inset map including ground zero signified by the star. Time series have varying y-axis scales and truncated x-axes.

similarly shallow water depth would transmit less energy into the water and towards wave-making. To rectify mass imbalance, the lip height could be lowered to better match the excavated volume.

## 5.3 Model implications

While re-emphasising this is a model case-use example, these preliminary results suggest some potential implications for modelling wave hazard from explosive subaqueous eruptions. If an eruption of sufficient magnitude at Lake Taupō produces an initial explosion there could be a threat posed to nearby shores. As with most lacustrine tsunami hazards, there is minimal time from source to shore impact; no existing warning system would ever be able to respond to an eruption with sufficient speed.

The underlying caldera has frequently experienced minor unrest (Potter et al., 2015), and current thought suggests eruption probabilities in the near-term are not negligible; for instance, an event of magnitude above that considered here ($0.1$ km$^3$) is estimated to have 5% probability within 100 years (Bebbington, 2020). Taupō volcano can produce far greater magnitude events than considered here, including the aforementioned CE 232 Taupō eruption (35 km$^3$ DRE) and the c. 25.5 ka Oruanui supere-

ruption (530 km³ DRE). Events of such magnitude, even when considering they may consist of multiple smaller episodes, undoubtedly carry wide ranging hazards well beyond the lake's proximity, and even those towards the lower end of the scale could produce numerous volcanic dangers, including ashfall and pyroclastic density currents. Therefore, an additional complexity of modelling the suite of hazards posed by subaqueous volcanism will be to determine the relative weight each source

component possesses with events of varying location and magnitude.

Only a brief effort is made at present on modelling this hazard as an example application of the multilayer numerical scheme and its benefits on resolving the resultant wavefield and significant outputs. Further work should incorporate this or similar numerical methods into conditional probabilistic hazard models for assessing the relative significance of this tsunami source mechanism, using wide parameter space comprising all likely sublacustrine eruption locations and magnitudes. Such a model

would be able to take advantage of this scheme's broad wave regime validity and computational efficiency, potentially able to investigate inundation in detail at various points onshore with, for instance, building data, key infrastructure impacts (e.g. control gates at outlet to the Waikato River), or similar additional model layers.

## 6   Conclusions

The non-hydrostatic multilayer scheme used in this paper has been shown to accurately replicate the collapse of various initial

disturbances into a resultant wavefield that exhibits varying degrees of non-linear properties and frequency dispersion. By capturing some depth aspects of the model without fully resolving the vertical, it is superior in accuracy to shallow water equation based schemes while being far more computationally efficient than direct numerical methods. The method was used to verify experimental results of positive amplitude disturbance wave generation and probe their validity at a wider parameter range while also further investigating negative amplitude disturbances, revealing extension of the leading trough relative to

trailing oscillations for larger size disturbances and smaller water depths.

Initialisations of wave generation via underwater explosion were tested for use with the multilayer scheme by simulating detonations of explosives as done in a US Army test series at Mono Lake. Including consideration of uncertainty with experimental data, the combination of empirically derived underwater explosion model and the numerical scheme was able to capture the significant elements of the generated waves as measured experimentally and help validate use of the underlying empirical

relations.

The multilayer scheme was then used to simulate an example volcanic explosion under Lake Taupō based on estimated eruptive energy by ejecta volume and MER. Implications for the developed model are suggested in the context of a small magnitude phreatomagmatic eruptions, with data revealing areas of varying exposure to waves, of above 0.1 m at locations near source, and waves reaching throughout the lake within 15 minutes. A probabilistic investigation is likely required to assess

the full range of possible scenarios at this location, including eruption geometry and size, while potentially considering further complexity, such as any syneruptive variation in initial conditions. In addition, any hazard model could usefully incorporate the numerical outputs in investigating cascading hazards such as one involving wave impact on the inlet dam at the Waikato river. Such a modelling effort would help resolve the significance of this hazard source compared to alternative tsunamigenic

sources and volcanic hazards across varying magnitudes of eruption. Further work could include numerical investigation of
non-detonation eruptions involving sustained jetting.

*Video supplement.*  Animation of the free-surface elevation of the Lake Taupō multilayer run is available from the TIB AV-Portal at https:
//doi.org/10.5446/52050 and for high quality on request.

*Author contributions.*  Conceptualisation, M.H., C.W., E.L., W.P.; Investigation, Validation, Formal analysis, Writing - original draft prepa-
ration, M.H.; Funding acquisition, Supervision, C.W., E.L., W.P.; Writing - review & editing, C.W., E.L., W.P., S.P., J.W

*Competing interests.*  The authors declare no conflicts of interest.

*Acknowledgements.*  We are grateful for the comments and suggestions of three reviewers, Stephan Grilli, Simon Barker and an anonymous
reviewer, which have markedly improved this work. This research was funded by the Marsden Fund Council, Royal Society Te Apārangi
grant number 17-NIW-017 awarded to NIWA. We wish to thank Sanjay Wadhwa for use of the bathymetric model of Lake Taupō courtesy
of MBIE/NIWA contract C01X1005 (Cumulative Effects of Stressors on Aquatic Ecosystems), and Waikato Regional Council for LiDAR
datasets of the Lake Taupō foreshore and surrounding topography. We acknowledge the use of New Zealand eScience Infrastructure (NeSI)
high performance computing facilities as part of this research.

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
