# Peer review of "Multilayer modelling of waves generated by explosive subaqueous volcanism"

_Natural Hazards and Earth System Sciences, 2021_

## Referee Comment (RC1)

**Reviewer's comments:**

**Major comments**

Intro: Mention other NH multi-layer models applied to dispersive and nonlinear tsunamis such as NHWAVE (e.g., Ma et al., 2012, 2013; Grilli et al., 2015, 2019, 2021; Schambach et al., 2019, 2020, 2021).

The present model is mentioned to have a good shoreline algorithm, but this is known to be a difficult problem for multilayer models unless the number of layers is gradually reduced towards shore. How is this done here ? How well can shallow water/nearshore results be trusted when many layers are used ? Please discuss and provide additional information.

The breaking criterion/dissipation of breaking wave energy represented by Eq (16) needs some support and/or physical justification. This is an important assumption that will affect the height of propagating tsunamis and bores.

L173: Please indicate papers in text where the scheme has been tested and validated.

L190: Prins' (1958) case appears to be relevant to the problem of concern here, although these are fairly small case experiments in which dissipation in breaking waves and through bottom friction may not be realistic or commensurate with field cases that are much larger cases (with much more turbulent flows). It would have been of interest to estimate the value of experimental Reynolds number and assess whether these were turbulent enough.

L205: for instance the breaking parameter is set to b=0.38 without justification. Is this to ensure a good agreement of model results with experiments ? Is this parameter general ? Would it be the same for breaking waves that are 100 m tall ? Is b dependent on ka and kh ? More support from earlier papers or justifications would be desirable here.

L229-230 Please indicate there are many phenomena neglected in the single phase multi-layer model used, and these might affect the level of dissipation. This also relates to the fact that in the explosion tests (in California), the model would overestimate generated waves if not for decreasing the explosion energy by 25% without a lot of justification for this value, except for a statement that energy released may have been smaller than nominal. Please discuss.

L226: One explanation for the strange results of the SGN model in the very near-field could be effects of very large vertical accelerations (ie non-hydrostatic pressure/dispersion) in the vertical column that are far outside the range of this model. Whereas in the far-field both ka and kh (not calculated by the way) would be back into acceptable ranges.

Fig. 5 is very interesting and important to understand the salient physics of this case. However, one is disappointed that kh and ka, the measures of dispersion and nonlinearity are not calculated nor discussed in the 2 field cases, with the former, if indeed very large (say beyond 3) justifying the need for a multi-layer NH model, rather than eg a SGN model.

The results discussed line 255-266 for positive or negative column, are closely similar to those obtained and discussed for positive or negative vertical bottom motions in experiments and model simulations (KdV), in their seminal papers, by Hammack (1973) and Hammack and Segur (1974, 1978a,b). Mention of this work and the similarity of physics and features in resulting wave trains would be interesting with a brief discussion.

Eqs. 1 and 2 and related parameters lack justification and appear to be simply stated here. Eq. (2) is used in the lake Taupo case but similarly without a justification. More explanations should be introduced at this stage in support of these equations.

L270-273: The text is not clear and somewhat misleading. One would understand depth to be 1946 m but then Fig 6 and earlier text mention 39 m and up to 45 m? Is 1946 m the lake MWL altitude ? If so, this really does not matter. And the sentence "… which left 2 m of shore topography.." should be clarified.

In this first field case as well as in the second one (and some earlier simulations) there is no mention of the horizontal grid range in the automatic refinement and the number of vertical layers, nor is there a convergence study justifying that the vertical discretization is sufficient. The model has automatic refinement but still some information on the numerical parameters used would be important to provide.

L276: The initial profile of the free surface is modeled with Eq. (2). This is stated without explanations or justification. Why not Eq (1). Were there field measurements indicating Eq. (2) was a good approximation ?

The Figures shown in Fig. 7 and 8 appear to have been inverted and do not correspond to the caption. Please correct.

In Fig. 8, the match of model and experiments requires a 25% reduction of the explosion energy. Besides the charges this could also reflect an inaccurate level of dissipation of breaking wave energy in the model in the near-field, also there could have been in the field some energy transferred to the bed as elastic deformation and elastic waves. Supporting insufficient breaking wave dissipation would be also the fact that later in the time series the model with reduced energy underpredicts experiments.\_Please discuss.

L311: Please replace or complement relatively deep nearfield by actual values of kh. There should be a discussion of nonlinearity and dispersion in the results shown here and in the next application. Also relate these values to those in Fig. 5 and hence the kind of wavetrain obtained.

L335: Like in earlier field case, some mention of the kh and ka values of computed wavetrains at gauges would be useful. It is pretty clear that a SV model will fail in this dispersive case but why not running the SGN model ?

In fact if one assumes depths of 20 or 50 m and periods of 15 or 65 s, one gets kh = 0.22 to 1.11 and for a = 3 m, ka = 0.01-0.15. So for the wavetrains at gauges, waves are moderately nonlinear and intermediate water so a SGN model should work well. Here as well no mention of the number of layers used in the NH model is made.

L359-362: As before, no information is provided on nb of layers required and since SGN was not tested one does not know if a NH multilayer model was really needed her, particularly in view of the large uncertainty on the initial empirical source shape and level of energy. Please discuss.

L343 and L356: the slower waves for the smaller V could be a result of reduced nonlinearity of wavetrains and hence amplitude dispersion effects. Please discuss.

L364: A SGN model such as e.g. FUNWAVE (Wei et al., 1995; Shi et al., 2012), which has extensively been applied and validated against tsunami benchmarks and case study (e.g., Watts et al., 2003; Day et al., 2005; Ioualalen et al., 2007; Abadie et al., 2012; Kirby et al., 2013; Grilli et al., 2015, 2019, 2021; Schambach et al., 2019, 2020, 2021; Tappin et al., 2008, 2014), particularly landslide tsunamis, also has NH pressure terms and breaking algorithm that have proved accurate in shallow water/nearshore. The multi-layer scheme may be needed for deep water explosions with very large kh values but not so much for the nearshore. This justifies many investigations of landslide tsunamis or tsunami from volcanic collapse, where a NH multilayer model was used in the near-field of the tsunami source and coupled to a SGN model for the far-field and runup/inundation where such models usually perform better than multi-layer ones. See, e.g., NHWAVE-FUNWAVE applications (e.g., Grilli et al., 2015, 2019, 2021; Schambach et al., 2014). A brief mention of this would be of interest at least in conclusions/discussions.

L380-381: Were the very large volumes listed here released at once in giant explosions or caldera collapses or were they the total deposits during a particular event. In this case only a small fraction could have been responsible for tsunami generation such as modeled here. Please be more nuanced in this statement.

L393-394: the actual nonlinearity and dispersion parameters were not mentioned nor discussed in the 2 field cases. This weakens the conclusions and the support for a multi-layer NH model.

**Minor comments**

L44 remove "is needed". L88 replace specialist by specialized ? L106 change to : where c=.. is an imperial... L212 lb of TNT ? Be specific Fig. 6a : Some contour lines of depth and topography would be useful. L267: replace charge by charge magnitude ? or energy ? L309: I would replace excellent by good or reasonable in view of the many hypotheses introduced to obtain a reasonable match between model and field data. Fig. 9: A table with actual location/depth of eruption and all the gauges would be useful. Information of gauge depth is mostly missing. In caption, replace building by built-up L341: relace all by the entire L340-343: text is not clear Fig. 10 caption. Please indicate this is for the larger V case. Fig. 11 caption: Make reference to table where gauge locations and depth are listed

**References**

Abadie, S., J.C. Harris, S.T. Grilli and R. Fabre 2012. Numerical modeling of tsunami waves generated by the flank collapse of the Cumbre Vieja Volcano (La Palma, Canary Islands) : tsunami source and near field effects. *J. Geophys. Res.*, 117, C05030

Day, S. J., P. Watts, S. T. Grilli and Kirby, J.T. 2005. Mechanical Models of the 1975 Kalapana, Hawaii Earthquake and Tsunami. *Marine Geology*, 215(1-2), 59-92, doi:10.1016/j.margeo.2004.11.008.

Grilli S.T., O'Reilly C., Harris J.C., Tajali-Bakhsh T., Tehranirad B., Banihashemi S., Kirby J.T., Baxter C.D.P., Eggeling T., Ma G. and F. Shi 2015. Modeling of SMF tsunami hazard along the upper US East Coast: Detailed impact around Ocean City, MD. *Natural Hazards*, 76(2), 705-746.

Grilli, S.T., Zhang, C., Kirby, J.T., Grilli, A.R., Tappin, D.R., Watt, S.F.L., Hunt, J.E., Novellino, A., Engwell, S.L., Nurshal, M.E., Abdurrachman, M., Cassidy, M., Madden-Nadeau A.L. amd S. Day 2021. Modeling of the Dec. 22nd 2018 Anak Krakatau volcano lateral collapse and tsunami based on recent field surveys: comparison with observed tsunami impact. *Marine Geology*, 440, 106566.

Hammack, J.L., 1973. A note on tsunamis: their generation and propagation in an ocean of uniform depth. *Journal* of *Fluid Mechanics*, 60(4), pp.769-799.

Hammack, J.L. and Segur, H., 1974. The Korteweg-de Vries equation and water waves. Part 2. Comparison with experiments. *Journal of Fluid mechanics*, 65(2), pp.289-314.

Hammack, J.L. and Segur, H., 1978a. The Korteweg-de Vries equation and water waves. Part 3. Oscillatory waves. *Journal of Fluid Mechanics*, 84(2), pp.337-358.

Hammack, J.L. and Segur, H., 1978b. Modelling criteria for long water waves. *Journal of Fluid Mechanics*, 84(2), pp.359-373.

Ioualalen, M., Asavanant, J., Kaewbanjak, N., Grilli, S.T., Kirby, J.T. and P. Watts 2007. Modeling the 26th December 2004 Indian Ocean tsunami: Case study of impact in Thailand. *Journal of Geophysical Research*, 112, C07024

Kirby, J.T., Shi, F., Tehranirad, B., Harris, J.C. and Grilli, S.T. 2013. Dispersive tsunami waves in the ocean: Model equations and sensitivity to dispersion and Coriolis effects. *Ocean Modeling*, 62, 39-55.

Ma, G., Shi, F. and Kirby, J.T., 2012. Shock-capturing non-hydrostatic model for fully dispersive surface wave processes. *Ocean Modelling*, 43, 22-35.

Ma, G., Kirby, J.T. and Shi, F., 2013. Numerical simulation of tsunami waves generated by deformable submarine landslides. *Ocean Modelling*, 69, 146-165.

Schambach, L., Grilli, S.T., Kirby, J.T. and F. Shi 2019. Landslide tsunami hazard along the upper US East Coast: effects of slide rheology, bottom friction, and frequency dispersion. *Pure and Applied Geophys.*, 176(7), 3,059-3,098

Schambach L., Grilli S.T., Tappin D.R., Gangemi M.D., and G. Barbaro 2020. New simulations and understanding of the 1908 Messina tsunami for a dual seismic and deep submarine mass failure source, *Marine Geology*, 421, 106093

Schambach L., Grilli S.T. and D.R. Tappin 2021. New high-resolution modeling of the 2018 Palu tsunami, based on supershear earthquake mechanisms and mapped coastal landslides, supports a dual source. *Frontiers in Earth Sciences*, 8, 627

Shi, F., J.T. Kirby, J.C. Harris, J.D. Geiman and S.T. Grilli 2012. A High-Order Adaptive Time-Stepping TVD Solver for Boussinesq Modeling of Breaking Waves and Coastal Inundation. *Ocean Modeling*, 43-44, 36-51

Tappin, D.R., Watts, P., Grilli, S.T. 2008. The Papua New Guinea tsunami of 1998: anatomy of a catastrophic event. *Natural Hazards and Earth System Sciences*, 8, 243-266

Tappin D.R., Grilli S.T., Harris J.C., Geller R.J., Masterlark T., Kirby J.T., F. Shi, G. Ma, K.K.S. Thingbaijamg, and P.M. Maig 2014. Did a submarine landslide contribute to the 2011 Tohoku tsunami ?, *Marine Geology*, 357, 344-361

Watts, S. T. Grilli, J. T. Kirby, G. J. Fryer, and Tappin, D. R. 2003. Landslide tsunami case studies using a Boussinesq model and a fully nonlinear tsunami generation model. *Natural Hazards and Earth System Sciences*, 3, 391-402.

Wei, J., Kirby, J.T, Grilli, S.T. and Subramanya, R. 1995. A Fully Nonlinear Boussinesq Model for Surface Waves. Part1. Highly Nonlinear Unsteady Waves. *Journal of Fluid Mechanics*, 294, 71-92.

---

## Referee Comment (RC2)

[referee-annotated manuscript omitted]

---

## Author Response (AR1)

This document includes responses to both reviewers. Changes made as a result of these discussions are included in an accompanying track changes document.

**Replies to comments of Reviewer #1:**

We thank the reviewer, Stephan Grilli, for the very detailed and thorough review and the effort taken over the comments. We are pleased to have found their suggestions exceptionally insightful, and our responses are as follows (in black), in order of the written comments (in blue):

Note: Some responses have been grouped to address similar comments together.

Intro: Mention other NH multi-layer models applied to dispersive and nonlinear tsunamis such as NHWAVE (e.g., Ma et al., 2012, 2013; Grilli et al., 2015, 2019, 2021; Schambach et al., 2019, 2020, 2021).

Reply 1:

Thank you for this suggestion; we shall include a brief on alternative multilayer models at the end of the paragraph at line 61 to contrast with the previous sentence (use of SWE equations for problem in focus) and for seamless tie-in with the following paragraph.

The present model is mentioned to have a good shoreline algorithm, but this is known to be a difficult problem for multilayer models unless the number of layers is gradually reduced towards shore. How is this done here ? How well can shallow water/nearshore results be trusted when many layers are used ? Please discuss and provide additional information.

The breaking criterion/dissipation of breaking wave energy represented by Eq (16) needs some support and/or physical justification. This is an important assumption that will affect the height of propagating tsunamis and bores.

Reply 2:

The present work focuses on initialisation of wavefields that exhibit varying degrees of non-linear properties and frequency dispersion and their propagation. Of the three cases, only the Taupō example quantifiably investigates shoreline effects (maximum crest heights at incidence and arrival times) and none investigate post-shoreline phenomena such as inundation. This itself is a focus of a further work in review external to this paper.

Details of the methods used for shoreline interaction and breaking with justification can be found in works by the scheme's primary author:

Sections 3.6.4 & 3.6.5 (Multilayer scheme):

Stéphane Popinet. A vertically-Lagrangian, non-hydrostatic, multilayer model for multiscale free-surface flows. *Journal of Computational Physics*, 418:109609, May 2020.

Section 3.4 (Methods used in SGN scheme):

Stéphane Popinet. A quadtree-adaptive multigrid solver for the Serre–Green–Naghdi equations. Journal of Computational Physics, 302:336–358, December 2015.

We appreciate that the description of the breaking algorithm is brief and, in addition to the description in Reply 5, propose adding the following referring text from Line 170:

"*…are sign and minimum functions respectively. The breaking algorithm used throughout the multilayer scheme is described in greater depth in the model's defining paper by Popinet (2020)*".

L173: Please indicate papers in text where the scheme has been tested and validated.

Reply 3:

The reference paper is indicated at the conclusion of the paragraph; this can be altered to in-text citation instead at Line 174:

"*…and breaking Stokes waves (Popinet, 2020). Source code of these examples can be found at http://basilisk.fr/src/layered/nh.h#usage.*".

Furthermore, it is important to provide suitable access to source code, as the typed hyperlink at line 174 provides permanent access. Many of the tests of the multilayer scheme were used previously for other schemes and, while often described well at source, are also described in more detail in these prior works and will also be referenced at this point:

S. Popinet. Quadtree-adaptive tsunami modelling. *Ocean Dynamics*, 61(9):1261–1285, 2011.
S. Popinet. Adaptive modelling of long-distance wave propagation and fine-scale flooding during the tohoku tsunami. *Natural Hazards and Earth System Sciences*, 12(4):1213–1227, 2012.
S. Popinet. A quadtree-adaptive multigrid solver for the Serre–Green–Naghdi equations. *Journal of Computational Physics*, 302:336–358, 2015.
S. Popinet. A vertically-Lagrangian, non-hydrostatic, multilayer model for multiscale free-surface flows. *Journal of Computational Physics*, 418:109609, 2020.

L190: Prins' (1958) case appears to be relevant to the problem of concern here, although these are fairly small case experiments in which dissipation in breaking waves and through bottom friction may not be realistic or commensurate with field cases that are much larger cases (with much more turbulent flows). It would have been of interest to estimate the value of experimental Reynolds number and assess whether these were turbulent enough.

Reply 4:

We thank the reviewer for this suggestion. We calculate the experimental Reynolds number across the whole parameter space for the initially generated wave to be in the range $7.2 \times 10^3$ - $3.2 \times 10^5$ when considering the depth averaged velocity and water height, with those towards the 'bore' regime demonstrating the highest as would be expected. It is worth reiterating that we are only using this validation case 'as is' for investigation across models for generating varying regime wave trains and do not end up scaling it up itself. Indeed, the Mono Lake example is another validation case at a larger scale.

L205: for instance the breaking parameter is set to b=0.38 without justification. Is this to ensure a good agreement of model results with experiments ? Is this parameter general ? Would it be the same for breaking waves that are 100 m tall ? Is b dependent on ka and kh ? More support from earlier papers or justifications would be desirable here.

Reply 5:

The 'breaking parameter' b is general and was not chosen to ensure good agreement of the model with experiments. Its value was instead used from the previous uses of the multilayer scheme in Popinet (2020). In Boussinesq-type models, such a parameter usually sets the steepness threshold whereafter the equations in use 'switch' to SWE to handle the breaking shock, for instance in works using Basilisk e.g. Beetham et al. (2016, 2018) and investigated well by Orszaghova et al. (2012). In this multilayer scheme, such an approach does not generalise easily and is provided by limiting the maximum vertical velocity based on the characteristic horizontal velocity scale, together with slope-limiting to ensure stability during stronger breaking. This approach we accept is a relatively simple parameterisation, and is validated with good success in Section 4.6 by Popinet (2020). In deep water, it is the same for breaking waves of varying height and while it may reflect ka, it does not depend on either ka or kh.

We propose that additional text and the following references will be added at Line 206 to justify the value used from prior work:

Beetham, E., Kench, P.S., O'Callaghan, J. and Popinet, S., 2016. Wave transformation and shoreline water level on Funafuti Atoll, Tuvalu. *Journal of Geophysical Research: Oceans*, *121*(1), pp.311-326.
Beetham, E., Kench, P.S. and Popinet, S., 2018. Model skill and sensitivity for simulating wave processes on coral reefs using a shock-capturing Green-Naghdi solver. *Journal of Coastal Research*, *34*(5), pp.1087-1099.
Orszaghova, J., Borthwick, A.G. and Taylor, P.H., 2012. From the paddle to the beach–A Boussinesq shallow water numerical wave tank based on Madsen and Sørensen's equations. *Journal of Computational Physics*, *231*(2), pp.328-344.
S. Popinet. A vertically-Lagrangian, non-hydrostatic, multilayer model for multiscale free-surface flows. *Journal of Computational Physics*, 418:109609, 2020.

L229-230 Please indicate there are many phenomena neglected in the single phase multi-layer model used, and these might affect the level of dissipation. This also relates to the fact that in the explosion tests (in California), the model would overestimate generated waves if not for decreasing the explosion energy by 25% without a lot of justification for this value, except for a statement that energy released may have been smaller than nominal. Please discuss.

Reply 6:

Thank you for this relevant comment; we will amend the sentence from line 231 to read:

"*…demonstrating the restriction of a single value for function on the 1D multilayer scheme not present on a 2D multiphase VOF solver, meaning that there are many neglected phenomena in the multilayer scheme such as bubbles and plunging breaks that, while this may be negligible in this lab-scale experiment, could be more significant at larger scale.*"

Further discussion about the Mono Lake energy disputation is included in Reply 14.

L226: One explanation for the strange results of the SGN model in the very near-field could be effects of very large vertical accelerations (ie non-hydrostatic pressure/dispersion) in the vertical column that are far outside the range of this model. Whereas in the far-field both ka and kh (not calculated by the way) would be back into acceptable ranges.

Reply 7:

We appreciate that little discussion was made on this observation and accept that a similar sentence can be added in explanation such as at line 228 (ka and kh are discussed in Reply 15):

"*…where, intriguingly, the water height temporarily increases above Q. This can be explained when considering the validity of these models does not extend to the very large vertical accelerations experienced near the shock, but instead out towards the resulting waves in the far-field.*"

The results discussed line 255-266 for positive or negative column, are closely similar to those obtained and discussed for positive or negative vertical bottom motions in experiments and model simulations (KdV), in their seminal papers, by Hammack (1973) and Hammack and Segur (1974, 1978a,b). Mention of this work and the similarity of physics and features in resulting wave trains would be interesting with a brief discussion.

Reply 8:

Thank you for this helpful observation, we will add the following paragraph after the one ending Line 258:

"*These results bear similarity to those found by Hammack and Segur (1974, 1978a,b) in experiments involving a piston producing vertical bottom motions described by Hammack (1973) and modelling using the Kortewig-de Vries equation, also with an initialised rectangular wave source. Notable similitudes include the generation of potentially multiple solitons of decreasing amplitude for shallow water positive initialisations (as seen in orange region of Fig. 5a), followed by a train of dispersive oscillatory waves and that no solitons are generated from negative vertical motions, instating producing a wave train of the type illustrated in Fig. 5b of an initial 'triangular' wave of greater speed than the trailing modulated oscillatory waves.*"

and adding the following to the reference list:

Hammack, J.L. and Segur, H., 1974. The Korteweg-de Vries equation and water waves. Part 2. Comparison with experiments. *Journal of Fluid mechanics*, *65*(2), pp.289-314.
Hammack, J.L. and Segur, H., 1978. The Korteweg-de Vries equation and water waves. Part 3. Oscillatory waves. *Journal of Fluid Mechanics*, *84*(2), pp.337-358.
Hammack, J.L. and Segur, H., 1978. Modelling criteria for long water waves. *Journal of Fluid Mechanics*, *84*(2), pp.359-373.

Eqs. 1 and 2 and related parameters lack justification and appear to be simply stated here. Eq. (2) is used in the lake Taupo case but similarly without a justification. More explanations should be introduced at this stage in support of these equations.

Reply 9:

These equations are introduced in Section 2.1 and are accompanied by a description of their derivation and referenced source, alongside the parameter definitions in Section 2.1.1 and a statement of the data sources used for calibrating the model (Le Méhauté and Wang, 1996). The equations themselves have been used in prior works of volcanic explosion context (Torsvik et al. 2010, Ulvrová et al. 2014, Paris & Ulvrová 2019), and this is mentioned in the opening paragraph of Section 2.1. We do agree with the reviewer that this justification is not clearly stated, and will add some discussion when introducing the equations.

While only one is used in this work, it is important to describe both to reflect that these are empirically calibrated models and alternatives that may possess variable validity were also produced by their authors. The choice of these used here reflects their prior use in previous work for similar contexts and its superior performance in the original deriving work; we understand that this is not sufficiently stated later in description of the field case methods so this can be primarily added into Section 4 and Line 275, and Section 5 will be amended to refer to the same methods used in Section 4.

L270-273: The text is not clear and somewhat misleading. One would understand depth to be 1946 m but then Fig 6 and earlier text mention 39 m and up to 45 m ? Is 1946 m the lake MWL

altitude ? If so, this really does not matter. And the sentence "... which left 2 m of shore topography.." should be clarified.

Reply 10:

Thank you for this observation, we will clarify the text starting at line 272 to the following: "*…where the lake water level was set at elevation above mean sea level 1945.7 m. This left approximately 2 m vertically of the bathymetric model dry to act as the shore surrounding the lake.*"

In this first field case as well as in the second one (and some earlier simulations) there is no mention of the horizontal grid range in the automatic refinement and the number of vertical layers, nor is there a convergence study justifying that the vertical discretization is sufficient. The model has automatic refinement but still some information on the numerical parameters used would be important to provide.

Reply 11:

While the parameter specifying maximum (horizontal) refinement level (plus resolution) and number of layers are specified in Section 3 (at end of 3.0 and in Table 1) and Section 5 (sentence beginning in Line 328), we acknowledge this information is absent in Section 4 and shall be included at a position such as at the end of the paragraph at Line 724.

L276: The initial profile of the free surface is modelled with Eq. (2). This is stated without explanations or justification. Why not Eq (1). Were there field measurements indicating Eq. (2) was a good approximation?

Reply 12:

As discussed in Reply 9, this will be amended.

The Figures shown in Fig. 7 and 8 appear to have been inverted and do not correspond to the caption. Please correct.

Reply 13:

Apologies for this oversight and thank you for the keen observation - this shall be corrected.

In Fig. 8, the match of model and experiments requires a 25% reduction of the explosion energy. Besides the charges this could also reflect an inaccurate level of dissipation of breaking wave energy in the model in the near-field, also there could have been in the field some energy transferred to the bed as elastic deformation and elastic waves. Supporting insufficient breaking wave dissipation would be also the fact that later in the time series the model with reduced energy underpredicts experiments. Please discuss.

Reply 14:

Thank you for these good points and we agree that they should be included in discussion in this section, for example as follows:

At Line 293:

"*The latter part of the initial wave group maintains higher amplitudes in the experimental trace for both records, whereas the envelope decay is sooner in the numerical model. Shot 3 also seems to exhibit a positive amplitude shift in the early part of the experimental envelope. These could be genuine underestimation of wave amplitudes towards the end of the initial group which could be due to variations in dissipation of the initially generated breaking waves.*"

At Line 303:

"*…in addition to the resultant early wave group and individual phases. Additionally, additional energy dissipation not accounted for in the physical model may be responsible for the greater fit of the reduced yield simulation, such as loses from a higher amount of dissipation from breaking of the initial waves from the explosion or some of the energy transferred to the nearby bed as elastic deformation. While the initialisation model is calibrated to charge depths relative to water depth, bed characteristics were not strongly considered.*"

Fig. 5 is very interesting and important to understand the salient physics of this case. However, one is disappointed that kh and ka, the measures of dispersion and nonlinearity are not calculated nor discussed in the 2 field cases, with the former, if indeed very large (say beyond 3) justifying the need for a multi-layer NH model, rather than eg a SGN model.

L311: Please replace or complement relatively deep nearfield by actual values of kh. There should be a discussion of nonlinearity and dispersion in the results shown here and in the next application. Also relate these values to those in Fig. 5 and hence the kind of wavetrain obtained.

L335: Like in earlier field case, some mention of the kh and ka values of computed wavetrains at gauges would be useful. It is pretty clear that a SV model will fail in this dispersive case but why not running the SGN model ?
In fact if one assumes depths of 20 or 50 m and periods of 15 or 65 s, one gets kh = 0.22 to 1.11 and for a = 3 m, ka = 0.01-0.15. So for the wavetrains at gauges, waves are moderately nonlinear and intermediate water so a SGN model should work well. Here as well no mention of the number of layers used in the NH model is made.

L359-362: As before, no information is provided on nb of layers required and since SGN was not tested one does not know if a NH multilayer model was really needed her, particularly in view of the large uncertainty on the initial empirical source shape and level of energy. Please discuss.

L364: A SGN model such as e.g. FUNWAVE (Wei et al., 1995; Shi et al., 2012), which has extensively been applied and validated against tsunami benchmarks and case study (e.g., Watts et al., 2003; Day et al., 2005; Ioualalen et al., 2007; Abadie et al., 2012; Kirby et al., 2013; Grilli et al., 2015, 2019, 2021; Schambach et al., 2019, 2020, 2021; Tappin et al., 2008, 2014), particularly landslide tsunamis, also has NH pressure terms and breaking algorithm that have proved accurate in shallow water/nearshore. The multi-layer scheme may be needed for deep water explosions with very large kh values but not so much for the nearshore. This justifies many investigations of landslide tsunamis or tsunami from volcanic collapse, where a NH multilayer model was used in the near-field of the tsunami source and coupled to a SGN model for the far- field and runup/inundation where such models usually perform better than multi-layer ones. See, e.g., NHWAVE-FUNWAVE applications (e.g., Grilli et al., 2015, 2019, 2021; Schambach et al., 2019, 2020, 2021; Tappin et al. 2014). A brief mention of this would be of interest at least in conclusions/discussions.

Reply 15:

Thank you for this very important and constructive comment. Following this and later comments, we will add results and discussion of the kh and ka type parameters to the two field cases. As the Taupō section as a whole is under revision following comments by Reviewer #2, the Mono Lake case additions only are described here:

In a new paragraph following Line 295:

"*In terms of parameters pertaining to dispersion and nonlinearity, kh and ka, waves in the initial group near the source at the nearest gauges on each radial were in the ranges 1.34 < kh < 3.56 and 3x10$^{-3}$ < ka < 1.181 and across the gauges beside the shore were in the ranges 0.105 < kh < 0.235 and 5x10$^{-4}$ < ka < 3.6x10$^{-3}$. As would be expected, moderately nonlinear waves are generated and kh decreases as the waves approach the shore and become shallow, whereas wave steepness ka decreases on average towards shore.*"

and at Line 311:

"*…accurately propagate such a source in the relatively deep (kh≈3) near-field through to the shore, it demonstrates suitability to model such events at these scales.*"

We acknowledge that while we used the SGN (Boussinesq-type approximation scheme) in the laboratory case, we did not perform the field cases with with it as we were primarily testing the multilayer scheme against a SWE method with the context of comparing against that most used for wave and tsunami modelling. We appreciate that the SGN method may produce results of similar contrast against SWE in this case, however our primary focus is that of demonstrating a usage case where a scheme capable of handling a wide range of dispersion characteristics as well as the initial source. Additionally, we found in initial investigations of both the cases in Section 3 and 4 that the multilayer scheme in Basilisk performed more efficiently than the SGN scheme in the same framework, for example included in this work in Table 1.

We propose adding multiple parts to address and inform the potential audience about the existence and capabilities of SGN-like models as described in the comments. Firstly following the paragraph ending Line 183, and secondly in an additional paragraph at the end of Section 5.1 after Line 369 which will detail alternatives to the multilayer scheme in this case, referencing ka and kh values with examples of SGN schemes like FUNWAVE and other higher schemes such as NHWAVE-FUNWAVE and SWASH.

L343 and L356: the slower waves for the smaller V could be a result of reduced nonlinearity of wavetrains and hence amplitude dispersion effects. Please discuss.

Reply 16:

Thank you for this point, we will include discussion of frequency and amplitude dispersion at these lines and pending the changes and corrections suggested by Reviewer #2 for this section.

L380-381: Were the very large volumes listed here released at once in giant explosions or caldera collapses or were they the total deposits during a particular event. In this case only a small fraction could have been responsible for tsunami generation such as modelled here. Please be more nuanced in this statement.

Reply 17:

If section is to be kept (considering Reviewer #2's suggestions), will consider rephrasing following sentence as:

"*Events of such magnitude, even when considering they may consist of multiple smaller episodes, undoubtedly carry wind ranging…*"

L393-394: the actual nonlinearity and dispersion parameters were not mentioned nor discussed in the 2 field ases. This weakens the conclusions and the support for a multi-layer NH model.

Reply 18:

 - As described in previous responses (Reply 15), these will be addressed such that this initial (or a rephrased similar) sentence is supported by the prior work.

**Minor comments**

L44 remove "is needed".
L88 replace specialist by specialized ?
L106 change to : where c=.. is an imperial...
L212 lb of TNT ? Be specific
Fig. 6a : Some contour lines of depth and topography would be useful.
L267: replace charge by charge magnitude ? or energy ?
L309: I would replace excellent by good or reasonable in view of the many hypotheses introduced to obtain a reasonable match between model and field data.
Fig. 9: A table with actual location/depth of eruption and all the gauges would be useful. Information of gauge depth is mostly missing.
In caption, replace building by built-up
L341: replace all by the entire
L340-343: text is not clear
Fig. 10 caption. Please indicate this is for the larger V case.
Fig. 11 caption: Make reference to table where gauge locations and depth are listed

Reply 19:

Fig 9: A table with shot and gauge location/depth information, available in the Mono Lake technical reports, will be included in an appendix or supplementary material.

L340-343: Rephrased as: "*The initial phase velocity typically starts from 40-45ms-1 until they heavily interact with the nearby Horomatangi Reefs. Due to the limited area of the lake, all of the shoreline experiences wave phenomena from these sources within 15 minutes, with only a slight increase of arrival times for the weaker run, due to both a slightly shorter period group generated by the smaller source and reduced nonlinearity.*"

All other minor suggestions accepted with thanks.

**Replies to comments of Reviewer #2:**

We thank the reviewer, Simon Barker, for their helpful and insightful review and generous comments on the work under review. We address the points raised (in blue) in order starting with the major points as written, followed by the minor comments (which were originally annotated on manuscript). Our responses are in black with suggested text revisions in red:

1) Estimating explosion energy. One of the hardest parameters to estimate as an input for the model is the energy release from the initial explosion of a volcanic eruption. A common approach is to use the crater size to estimate total energy release. This is a simplification, but has been a useful in some cases (e.g. Taal, where there are explosion craters to measure and historic eruptions for comparison: Paris and Ulvrova, 2019). However, volcanic eruptions are often much more complex. Particularly those from rhyolitic calderas. In the case of the Taupō scenario used here, the authors have chosen eruptive volumes of 0.04 and 0.4 km3 and then back-calculated the theoretical crater size and hence energy release from such an explosion. However, these volumes from past eruptions are from whole eruption sequences that may have occurred over hours to days. Using these volumes to represent a single explosion is not a wise approach and probably leads to an overestimation of energy that has little volcanic or geological context. See more detailed comments in the manuscript.

Reply 1:

Thank you for this important point. While we emphasise that the inclusion of this volcanic example is mainly for a demonstrative purpose and not to act as a full hazard assessment, we agree that more care needs to be taken in the treatment of the source and its justification. We propose changing the modelling scenarios in this last section as follows:

 - Removing the two source sizes (0.04 & 0.4 km³) investigated presently.
 - Adding modelling of one source size derived from an approximation of mass eruption rate (MER) from the literature for a moderate size eruption at Taupō, for example the following parameters which is then input into the existing explosion model:

| MER (kg s⁻¹) | Simulation Time (s) | Ejecta Volume DRE (km) |
|---|---|---|
| 1.2E+07 | 1000 | 4E-03 |

This involves re-running of the model with these parameters, its insertion into the section and adjustment of the figures accompanying it, which is relatively straightforward and preliminary results from this in the form of an equivalent Figure 10 are presented below. The results obtained reflect that of a much smaller source in that wave amplitudes are significantly smaller, especially near-source, and the speed of propagation of these waves across the lake is slower. Discussion of the results of this revised example will involve many of the same components as originally done, with addition of additional discussion of the wavefield as suggested by Reviewer #1, by complementing the early work in the manuscript with highlighting ka and kh in this example and relating it to results in Sections 3 and 4.

[Figure]

Figure 10. Multilayer simulation of potential volcanic explosion in Lake Taupō. (a) First arrival travel times in minutes. (b) Field of maximum crest amplitudes. (c) Snapshot of wave amplitudes at t = 4.5 minutes. (d) Equivalent of c) but using the Saint Venant scheme.

2) Little discussion over background volcanology. Scenarios for eruptions from Lake Taupō have little background other than saying they fall broadly within an area where eruptions have occurred over the past 5000 years. The references cited in this section are very sparse and volumes and eruption ages are often given without correct citation. There is very little discussion around where the most recent vents were, eruption styles inferred from the geological record, or where the current areas of hydrothermal venting are. Surely a phreatic explosion would also be a good scenario to model? The depth may also be highly variable, but little attention is given to why the particular depth used was chosen.

In summary, I think that the manuscript has the potential to be valuable for refining tsunami modelling and for application to subaqueous explosions. However, the illustrated use of the model and the scenario chosen for Taupō does not seem to be constrained by the appropriate data and therefore likely generates results that have little context for hazard assessment. I would therefore suggest that either the scenario for Taupō needs to be more carefully refined with justification behind the parameters used, or that it be dropped from the current manuscript to allow the focus of the paper to be solely on the new multilayer scheme for wave generation.

Reply 2:

Thank you again for the detailed review. As stated in Reply 1, we would intend to revise the parameters behind the Taupō example (Section 5) using a slightly refined method enabling additional justification. In addition to this, in order to satisfactorily complement this change and provide additional discussion over background volcanology, we propose to:

 - Add introduction paragraph to Section 5 about how we now apply the multilayer model to a volcano as an example. Emphasise it's to demonstrate use, not hazard assessment.

 - Include further background volcanology in the paragraph following, including typical eruptive styles and range of sizes, most recent events and general locations, and most recent hydrothermally active areas.

 - Provide additional justification towards the chosen source size and location. Size as described in Reply 1, and location complemented by previous bullet describing the most recent events and hydrothermal features.

 - Section 5.2 (Hazard implications) will be renamed to 'discussion' and be re-focused away from explicitly discussing impacts on hazard modelling, instead discussing these but primarily with model implications. It will be continually stressed that this is a use-case example of the model focussed on earlier in the manuscript and that many assumptions are required in modelling this scenario, leading to concluding that lots of care would be needed in specifying (and then justifying) a source model and its assumptions if utilising these methods for future work in a more complete hazard study.

 - Reword conclusions (from Sect. 5, Line 405) to reflect more that this is an example - will focus on the modelling outcomes rather than hazard outcomes.

Reply 3:

Most comments as annotated on the manuscript are addressed as part of Replies 1 and 2. The remaining are detailed and answered below:

Abstract, Line 9:  I think this needs to be reworded given the comments below over relationships between eruption volume and initial explosion. You also don't model a range between these end members. Reword.

Reply 4:

Text within Lines 9-11 shall be reworded as part of the planned revisions detailed in Replies 1 and 2.

Line 20:  This sentence is a little awkward as written. Are you trying to say that they kill more people than other volcanic hazards? How does 5% of tsunamis being volcanic relate to the number of fatalities from volcanoes? These seem like 2 different concepts mixed up.

Reply 5:

Will reword as follows:

"*While volcanoes are estimated to be responsible for just 5% of all noted tsunamis since 1600 AD, they can be particularly dangerous in that they account for 20-25% of all recorded fatalities resulting from volcanic activity.*"

Line 67:  This would also be a good place to note that you do not do a full probabilistic hazard assessment but that will be the focus of another paper.

Reply 6:

Text will be added at this point to reflect this:

"These tests are to establish fitness of the underlying models, which are then applied to a hypothetical explosive submarine eruption at Lake Taupō, New Zealand as an example use case in hazard applications."

Line 76:  here and elsewhere. You switch between subaqueous and submarine several times. Perhaps stick with subaqueous for general application and then only use submarine when talking about examples in the sea?

Reply 7:

In the context of the cases and locations used in this paper, we agree it is more suitable to consistently use 'subaqueous' primarily the work, leaving subaqueous to actual marine cases discussed early on (and additionally 'laucustrine' where necessary in Sections 4 and 5). This will be revised throughout:

 - Title
 - Abstract Lines 2, 4, 12
 - Lines 16, 41, 55, 76, 191, 310, 383
 - Fig. 1 caption

Line 299:  not clear why this data is not shown? Does it not exist? just need some clarification

Reply 8:

Unlike the series in 1965, the data from the three additional shots in 1966 were only partially included in the technical reports on the series as it seems they were primarily focused on investigating run-up and other phenomena rather than the immediately generated wavefield. We will refine this sentence to read:

"Further similar deep tests at Mono Lake in the following year, which were conducted for other investigations, delivered much greater amplitude waves in line with expectations, leading to the suggestion by Wallace and Baird (1968) that…"

Line 312:  and note that these timescales are suitable for very short lived events. What would be useful below before jumping into the taupo scenarios is to discuss the limitations of the model and applicability to volcanic eruptions in general

Reply 9:

A small rephrasing will be made at Line 311 to address this:

"…it demonstrates suitability to model such events at these spatial and time scales. However, this test did not contain… any shallow depth underwater explosion tests, for which there are no case examples at this scale."

---

## Referee Report (RR1)

**General comments:**

This revised manuscript illustrates a numerical method aimed to reproduce the tsunami waves generated by explosive subaqueous volcanism. The proposed numerical model is based on a non-hydrostatic multiphase solver for free-surface flows within the CFD code - Basilisk. The manuscript details the numerical validation against large-scale experimental data and field observations. The model is then applied to simulate a hypothetical case study (Lake Taupō), demonstrating its potential to be used for the relevant hazard assessment. I found the manuscript interesting and it is my opinion the topic perfectly fits the aim of the NHESS. Nevertheless, I kindly ask the authors to address some minor comments before I can recommend the manuscript for final publication.

**Minor comments:**

Section 4: I think it would be good to add a brief description about Shots 3 and 9. It is not clear where they are from.

L44 and L64: Please check the order of the references, as they are currently not in alphabetic or chronological orders.

L205: Please give the full name of "NH" first (probably on L61) and then use its abbreviation.

L242: In the entire manuscript, you switch between "Fig." and "Figure" several times. Please make them consistent.

L329 and L330: Physical variables should be written in Italic to be consistent with other places in the manuscript.

L492: The issue number C5 is missed.

References list: A careful revision of this part is recommended. I noticed that some references are given with the DOIs but others are not. It would be good to try to give the DOIs for at least all the peer-review journal articles. Furthermore, please give the full lists of authors for some references (e.g. on L545, L548, L571).

---

## Author Response (AR2)

**Replies to reviewers' comments:**

We thank the reviewers for the additional reviews requested and the effort taken over them. For both reports, of which we have accepted all suggestions, our responses are as follows (in black), in order of the written comments (in blue):

Section 4: I think it would be good to add a brief description about Shots 3 and 9. It is not clear where they are from.

Reply 1:

Thank you for this suggestion; we shall include a brief description as such at L304:

"*Data from shots 3 and 9, individual explosive tests within the experimental series, are chosen to test the numerical method against.*"

L44 and L64: Please check the order of the references, as they are currently not in alphabetic or chronological orders.

Reply 2:

This has been resolved now in the LaTeX manuscript for these instances. See also reply ???

L205: Please give the full name of "NH" first (probably on L61) and then use its abbreviation.

Reply 3:

Thank you for this technical correction, this has been added at the point specified.

L242: In the entire manuscript, you switch between "Fig." and "Figure" several times. Please make them consistent.

Reply 4:

Thank you, we have chosen to change all instances to "Figure" throughout the manuscript.

L329 and L330: Physical variables should be written in Italic to be consistent with other places in the manuscript.

Reply 5:

Thank you, this has been rectified.

L492: The issue number C5 is missed. References list: A careful revision of this part is recommended. I noticed that some references are given with the DOIs but others are not. It would be good to try to give the DOIs for at least all the peer-review journal articles. Furthermore, please give the full lists of authors for some references (e.g. on L545, L548, L571).

Reply 6:

Thank you for noticing this; it seems that Copernicus' provided bibtex style is the issue with limiting printing full author lists over some specified limit - however the full author lists are indeed in the manuscript file. We have also adjusted and cleaned the bibtex entries to ensure that all DOIs (where available) are printed and shown, as well as issue numbers and volume numbers. A further issue was some middle initials being skipped which has also been solved.

Line 9: "Magnitude range" - you only present 1 example now. Remove range. It is also a little confusing to state total eruption volume here when you are only modeling 1000s using a mass eruption rate that might be expected for this size. Perhaps replace eruption volume with eruption rate?

Reply 7:

Thank you for these suggestions, we will remove "range" from this line as it was an oversight. However, we insist that inclusion of the modelled equivalent ejecta volume is more pertinent in the abstract than the eruption rate as, in lieu of any other comparison which is not made in this work, a comparison against historical eruptions is the most relevant.

Line 360: "Taupo volcano offers great opportunities for investigating post-supereruption magmatic system..." You are not investigating the magmatic system. Reword. Actually you could probably reduce the amount of information in the following paragraph given that you only now model 1 eruption size.

Reply 8:

We shall rephrase from L360 to:

"Taupō volcano and its post-supereruption magmatic system have been extensively studied because of how recently the event occurred, along with the volcano's relatively high activity. As a result…"

Line 367: Volume of dacites. Check these numbers and state if bulk or DRE (dense rock equivalent) magma. Should be 0.01 to 0.1 volume bulk (not magma). Also check for rhyolites on line 369.

Reply 9:

To keep consistency with modelled values, we shall remove inconsistencies and prefer to use DRE. We will define the acronym at L368, and use it at the instances at L368, L370, L445 and L446.

Line 373: Horomatangi Reefs were technically post-Taupo eruption by 20-30 years. See Barker et al. (2016). doi:10.1130/G37382.1

Reply 10:

Thank you for this point; we shall amend the end of this paragraph as such:

"…and resulted in the further collapse of the caldera beneath the lake and, afterwards, the formation of the Horomatangi Reefs (Davy and Caldwell, 1998)."

Line 383: You should finish off this paragraph with something like "Here we model a single eruption from Taupo to demonstrate the multilayer model, but highlight that future work will be required to provide an assessment of tsunami hazards from this volcano".

Reply 11:

Thank you - we shall accept this suggestion as it is written.

Line 397: "Endure for hours to days" Reference?

Reply 12:

The following reference has been added at this point:

Pyle, D. M.: Chapter 13 - Sizes of Volcanic Eruptions, in: The Encyclopedia of Volcanoes (Second Edition), edited by Sigurdsson, H., pp. 257–264, Academic Press, Amsterdam, second edition edn., https://doi.org/10.1016/B978-0-12-385938-9.00013-4, 2015.

Line 405: "Larger simulation": You only show 1 example now!

Reply 13:

This oversight has been removed, thank you.

Line 444: Taupo eruption volume: Should be 35km3 magma or ~105km3 bulk. Note that the 0.1km3 you state for the smaller size

Reply 14:

Thank you, this has been amended. Please see also Reply 9.

Line 455: This is a great way to finish the paper now.

Reply 15:

Many thanks for the kind comments.

Line 469: Replace "These were then used" with "The multilayer scheme was then used...

Reply 16:

This has been changed as written.

References: These are still a bit of a mess. I've noticed on quite a few papers (just of mine) that people with 3 initials have been reduced to 2 and sometimes 1. E.g. C.J.N. Wilson (sometimes C. Wilson), A.S.R Allan etc just to name a few I know. Some papers have DOI's, others do not. Some journal names are in caps, some are all lowercase. Check throughout.

Reply 17:

Please see Reply 6 for the response and resolution to these issues.